



Geoscientific
Model Development

# Accounting for forest management in the estimation of forest carbon balance using the dynamic vegetation model LPJ-GUESS (v4.0, r9710): implementation and evaluation of simulations for Europe

Mats Lindeskog[1], Benjamin Smith[1,2], Fredrik Lagergren[1], Ekaterina Sycheva[3], Andrej Ficko[4], Hans Pretzsch[3], and Anja Rammig[3]

[1]Department of Physical Geography and Ecosystem Science, Lund University, Lund, Sweden
[2]Hawkesbury Institute for the Environment, Western Sydney University, Penrith NSW, Australia
[3]TUM School of Life Sciences Weihenstephan, Technical University of Munich, Freising, Germany
[4]Biotechnical Faculty, University of Ljubljana, Ljubljana, Slovenia

**Correspondence:** Mats Lindeskog (mats.lindeskog@nateko.lu.se)

**Abstract.** Global forests are the main component of the land carbon sink, which acts as a partial buffer to $CO_2$ emissions into the atmosphere. Dynamic vegetation models offer an approach to projecting the development of forest carbon sink capacity in a future climate. Forest management capabilities are important to include in dynamic vegetation models to account for the effects of age and species structure and wood harvest on carbon stocks and carbon storage potential. This article describes the implementation of a forest management module containing even-age and clear-cut and uneven-age and continuous-cover management alternatives in the dynamic vegetation model LPJ-GUESS. Different age and species structure initialisation strategies and harvest alternatives are introduced. The model is applied at stand and European scales. Different management alternatives are applied in simulations of European beech (*Fagus sylvaticus*) and Norway spruce (*Picea abies*) even-aged monoculture stands in central Europe and evaluated against above-ground standing stem volume and harvested volume data from long-term experimental plots. At the European scale, an automated thinning and clear-cut strategy is applied. Modelled carbon stocks and fluxes are evaluated against reported data at the continent and country levels. Including wood harvest in regrowth forests increases the simulated total European carbon sink by 32 % in 1991–2015 and improves the fit to the reported European carbon sink, growing stock, and net annual increment (NAI). Growing stock

($156 \, \mathrm{m}^3 \, \mathrm{ha}^{-1}$) and NAI ($5.4 \, \mathrm{m}^3 \, \mathrm{ha}^1 \, \mathrm{yr}^1$) densities in 2010 are close to reported values, while the carbon sink density in 2000–2007 ($0.085 \, \mathrm{kg} \, \mathrm{C} \, \mathrm{m}^{-2} \, \mathrm{yr}^1$) equates to 63 % of reported values, most likely reflecting uncertainties in carbon fluxes from soil given the unaccounted for forest land-use history in the simulations. The fit of modelled and reported values for individual European countries varies, but NAI is generally closer to reported values when including wood harvest in simulations.

## 1 Introduction

Forests globally provide ecosystem services including provision of timber, fuel, and water; regulation of local climate and hydrology; carbon sequestration; support of biodiversity; and recreation (Bonan, 2008; Mori et al., 2017). The effects of climate change on forest productivity and biodiversity may be predicted to be negative due to increased evapotranspiration and reduced rainfall in many forested areas; an increase in extreme events like droughts, wildfires, storms, and insect attacks; and local or regional extinctions of plant and animal species (Easterling et al., 2000; Seidl et al., 2011; Anderegg et al., 2013; Urban, 2015). On the other hand, productivity may increase due to the fertilising effect of increased nitrogen deposition and higher atmospheric $CO_2$ levels (Zaehle and Dalmonech, 2011; Luyssaert et al., 2008) and shifts in

tree species composition and longer growing seasons at high latitudes caused by higher temperatures (Sitch et al., 2015; Morin et al., 2018).

Forests make up the largest portion of the current land carbon sink and are estimated to have absorbed 20 %–50 % of $CO_2$ emitted by fossil fuel combustion and industry during the first decade of this century (Pan et al., 2011; Le Quéré et al., 2018; Pugh et al., 2019). The suggested basis for this carbon uptake is the recent history of the drivers increasing productivity mentioned above, especially increased $CO_2$, and the recovery of carbon pools in regrowth forests (forests regrowing after natural or anthropogenic stand-destroying disturbances; Pugh et al., 2019). The size of the forest carbon sink has been estimated by using bookkeeping methods (Pan et al., 2011; Houghton et al., 2012) and global vegetation models (Luyssaert et al., 2008; Shevliakova et al., 2009; Pugh et al., 2019), but this sink is associated with relatively large uncertainties, resulting in differing estimates using different approaches and models. Key uncertainties include the magnitude of $CO_2$ fertilisation – which may be limited by soil availability of nutrients such as N and P (Zaehle and Dalmonech, 2011; Jiang et al., 2020) – and the extent of shifting cultivation in the tropics (Heinimann et al., 2017). While the net atmosphere-to-land flux ($F_L$) is relatively well constrained by atmospheric measurements, large uncertainties in the net land-use and land-cover flux ($F_{LULCC}$) make the size of the residual (land) sink ($F_{RL}$) itself uncertain ($F_L = F_{RL} - F_{LULCC}$) (Arneth et al., 2017).

Forests cover 33 % of Europe's land area (Forest Europe, 2015) and store approximately 13 Pg C in vegetation and 28 Pg C in soils (Pan et al., 2011). The carbon sink of European forests in 2000–2007 has been estimated at 0.27 Pg C yr$^{-1}$ or about 12 % of the global carbon sink of established forests (Pan et al., 2011). Europe has been identified as a region where regrowth forests dominate carbon sequestration (Pugh et al., 2019) and has a history of thousands of years of human impact on forest structure and species composition (Perlin, 2005). Forest management practices of the past few hundred years are relatively well documented (McGrath et al., 2015). Depending on the region, different management strategies are applied (Cardellini et al., 2018). The preponderance of young trees and the removal of wood in managed forests influence carbon stocks and fluxes, e.g. by increasing productivity and reducing self-thinning, age-related mortality, and litter production compared to pristine forests (Zaehle et al., 2006). In addition to the effects on radiative forcing by atmospheric $CO_2$, forest management influences local climate by changing albedo, evapotranspiration, and surface roughness (Luyssaert et al., 2014).

Dynamic vegetation models (DVMs) provide a potential framework for predicting the combined effects of climate and forest management scenarios on forest ecosystem structure and carbon balance. Based on such information, the potential of forest landscapes to contribute to climate change mitigation by maintaining or enhancing carbon sinks and to climate adaptation through sustained production of forest products and other ecosystem services in the face of climate change can be assessed. Applications of DVMs to represent climate responses of potential natural vegetation (PNV) have been shown in the past, for example as a basis for nature conservation planning (Hickler et al., 2012). Human management of land, including cropland, pasture, and managed forest, has been introduced in a number of global DVMs (Bondeau et al., 2007; Bellassen et al., 2010; Lindeskog et al., 2013; Arneth et al., 2017). Key elements required to represent managed forests in a DVM framework include the ability to initialise a simulation with historical land use; to represent age and size structure of forests stands and their change over time; to account for tree species composition; and to apply silvicultural treatments that modify stand composition and structure like planting, thinning, and harvesting.

LPJ-GUESS (Smith et al., 2001, 2014) is a second-generation DVM tailored for regional- and global-scale applications. It is one of few globally applicable DVMs that incorporate a detailed representation of forest ecosystem composition and stand dynamics, suitable for the implementation of a forest management scheme. It captures the distribution of European PNV at species level and can make projections of vegetation shifts under future climate scenarios (Hickler et al., 2012). The model has been shown to represent stand-level vegetation growth and succession successfully (Smith et al., 2014). It has been used to estimate forest vulnerability to climate change (Seiler et al., 2015) and carbon mitigation potential of regrowth forests and forests under alternative management scenarios (Pugh et al., 2019; Krause et al., 2020). Earlier versions of LPJ-GUESS have been modified to enable analysis of clear-cut forest management and the effects of wind damage and insect outbreaks (Lagergren et al., 2012; Jönsson et al., 2012). In this study, we describe the implementation of expanded forest management capabilities including even-age and clear-cut as well as uneven-age and continuous-cover management in LPJ-GUESS v4.0. In addition to detailed carbon and water cycle processes, this version of the model incorporates a dynamic nitrogen cycle and nitrogen limitation on plant productivity (Smith et al., 2014). In this way, forest management in LPJ-GUESS is for the first time fully integrated in a model version capable of simulating a landscape containing a mosaic of land-cover types like PNV, cropland, pasture, and peatland and with a sophisticated land-use and land-cover change functionality. Model alternatives for forest stand initialisation (land-use history and species and age distribution) and silvicultural management (detailed and automated harvest strategies) are presented in detail. Simulations using different forest management alternatives are evaluated against observations of standing volume and harvest for even-aged monospecific European beech and Norway spruce stands in central Europe. Using an automated thinning and clear-cutting approach for European forests, we compare modelled carbon stocks and

fluxes with observational data and explore the dynamic behaviour of the model under changing climate forcing.

## 2 Methods

### 2.1 General description of LPJ-GUESS and overview of simulated processes

LPJ-GUESS (Smith et al., 2001, 2014) simulates the dynamics of terrestrial vegetation and soils across a regional or global grid, forced by meteorological and land-use inputs and soil physical properties. In the absence of land use, each grid cell encompasses a landscape of natural, climatically determined vegetation (PNV). Replicate patches, nominally 0.1 ha ($1000\,\mathrm{m}^3$) in size, represent disturbance-related variation in stand age across the wider landscape of a grid cell. In each patch, age cohorts of tree plant functional types (PFTs) or species and shrub and grass PFTs compete for light, water, nitrogen, and space (Fig. 1). Photosynthesis, respiration, phenology, soil carbon and nitrogen cycling, and hydrology occur at a daily time step, while biomass growth allocation, turnover, establishment, and mortality occur at a yearly time step. In its original version, the model only simulated PFTs that capture the major vegetation zones globally. The parameter set of these PFTs has been extended to simulate the most important tree species in the north-eastern USA (Hickler et al., 2004) and Europe (Koca et al., 2006; Hickler et al., 2012) as distinct PFTs. The new functionality defined in this paper can operate equally on individual tree species or more generalised PFTs. Hereinafter "species" is thus used synonymously with "PFT". The forest canopy is represented as a multi-layered structure. Leaves, fine roots, and stem heartwood and sapwood are represented as dynamic pools for each age cohort of each PFT. Branches and course roots are not explicitly discriminated but are implicit in the wood biomass pool. The patches are subject to stochastic vegetation-destroying disturbance events (representing, e.g. wind storms or landslides) with a prescribed return time (e.g. 100–400 years). Disturbance results in the loss of vegetation in a patch, after which a secondary succession of grass and tree PFTs follows (Hickler et al., 2004). Establishment is affected by forest floor light conditions and is subject to PFT-specific environmental envelopes defined by bioclimatic limits. A slightly different set of bioclimatic limits govern survivorship (Table A1). Mortality resulting from self-thinning, reduced growth efficiency, old age, and wild fires are applied to individual cohorts. Establishment and mortality have a stochastic component. Soil carbon and nitrogen cycling are based on the CENTURY model (Parton et al., 1993), and soil hydrology is based on a two-layered "leaking bucket" model. A soil mineral nitrogen pool is provided by atmospheric deposition, biological nitrogen fixation and gross nitrogen mineralisation of soil organic matter. Plant nitrogen uptake is driven by the demand from photosynthesis and biomass growth and is limited by the supply from the soil mineral nitrogen pool. The nitrogen cycling scheme is described in detail by Smith et al. (2014).

Different land-use and land-cover types in addition to PNV are represented in the model by stand types with different management, e.g. cropland, pasture, and managed forest (Lindeskog et al., 2013, Fig. 1). Transitions between different stand types may occur at any point in time, according to land-use data inputs, to take into account land-use history or future land-use scenarios. When a potentially forested stand type area expands, new stands are created, keeping the soil history from the previous stand type intact and allowing vegetation succession to proceed from bare ground (in most cases; see Sect. 2.2.1). In modelled wood harvest events, 66 % of wood biomass and 30 % of leaf biomass are typically removed from the stand and the rest remains as litter. Removed leaf biomass and part of wood biomass (by default 67 %) is oxidised the same year. The remaining wood biomass is put into a product pool with a turnover rate of 4 % per year.

The typical forest management types covered in the model and presented in this paper are no management (pristine forests, simulated as PNV), even-aged forestry, typically modelled by stands with prescribed ages starting from bare ground after a specified land-use history, and uneven-aged/continuous cover forestry, typically modelled by a cohort structure within a patch derived from prescribed cuttings after starting as bare ground and a regeneration phase. Alternatives to these typical setups can be used to achieve age structures at other spatial scales, e.g. landscape level, and will be described below.

### 2.2 Forest structure initialisation routines

Forest stand age and species distributions can be achieved in the model by utilising the structure of a previous PNV stand or by defining a new age and species structure at various levels of detail.

#### 2.2.1 Stand creation

A managed forest stand may be created in the model by two different options (Figs. 2, B1). By cloning the parent stand, the complete state with all patches intact is inherited by the secondary stand. If the origin is previous woodland (PNV or secondary forest), a cutting scheme may start with the existing tree structure, optionally cutting unwanted species. In the other alternative, tree growth starts from bare ground after an initial clear-cut or when expanding on former cropland or pasture. In this case (with an even-age stand and if disturbance and fire are turned off), the secondary stand can in many cases be modelled by a smaller number of replicate patches since there is usually no random variation in the timing of management events.

https://doi.org/10.5194/gmd-14-1-2021      Geosci. Model Dev., 14, 1–42, 2021

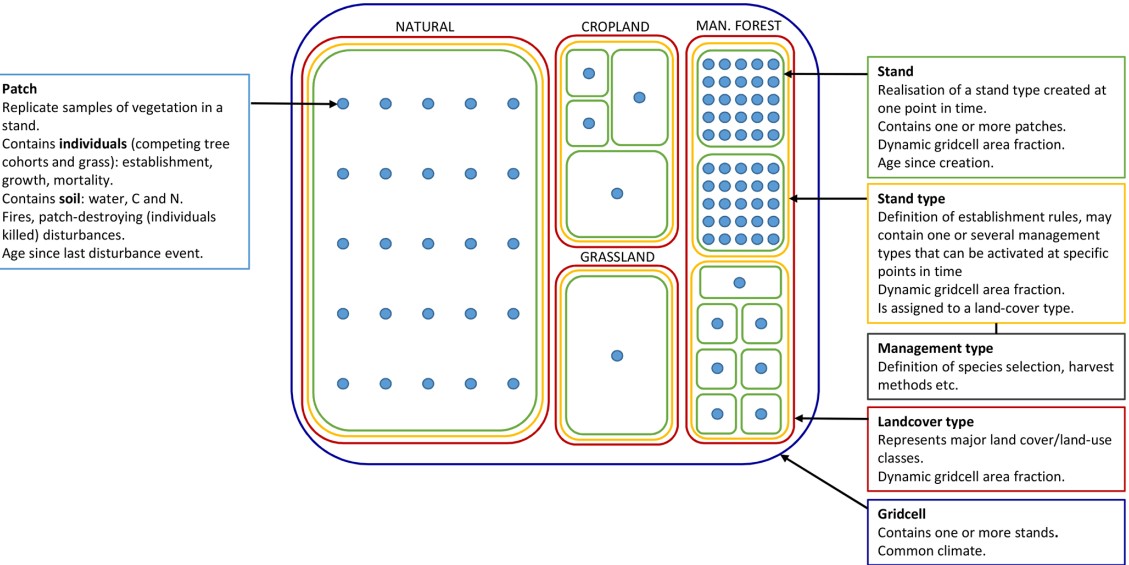

**Figure 1.** Data structures in LPJ-GUESS relevant for this study. Patch number is defined separately for PNV and secondary stands. If a secondary stand is created from PNV or managed forest with intact vegetation, the patch number of the parent stand is used. During land-cover change events, stands belonging to forest stand types can only be reduced in size. Expansion of such stand types results in new stands.

### 2.2.2 Secondary forest age structure

Managed forest stands with an uneven age structure can be represented in the model by selecting different options, depending on the spatial context of the age classes, i.e. whether they correspond to tree age cohorts co-occurring within local stands thereby competing with each other or represent different fractions of a wider landscape with no local interactions between age cohorts. An age structure may be created in individual patches by thinning (enabling regeneration by increased light at the forest floor) at defined intervals during an initialisation period, allowing for both intraspecific and interspecific competition (Fig. 3a). When competition between different age classes does not apply, i.e. when the spatial context is that of a landscape, different age-classes can be modelled in separate patches. To achieve an age structure among patches within a stand, the semi-randomised age structure of PNV (see Sect. 2.1) may be conserved after the conversion to managed forest if the cloning functionality is used (Fig. B1). Alternatively, multiple patches in a secondary stand may be clear-cut successively, one by one, at regular intervals during an initialisation period (Fig. 3b). In the final approach, a prescribed age structure, either representing a specific moment in time, or a historical development, may be created among stands representing a stand type using land-cover change input data (Fig. 3c).

### 2.2.3 Secondary forest species composition

Species mixtures may be defined either at the management type level (Fig. 1), using predefined planting densities for individual species and/or later cuttings to achieve prescribed relative biomass abundances of the different species within a patch (Fig. 4a, see below), or at the landscape level, using land-cover input data to achieve predefined groundcover area-based mixtures of monocultures (Fig. 4b), or a combination of both of these options.

### 2.3 Forest management routines

Two types of harvest systems are available in the model: clear-cutting and continuous cutting, which are used in conjunction with the even-aged and uneven-aged/continuous-cover age structure systems, respectively (Table 1). Depending on the level of detail in historic forest management input data or, in simulations of future scenarios, whether the management should be able to adapt to a changing climate or other factors, various model alternatives are available.

### 2.3.1 Simplified clear-cut forestry

A simplified method to represent forestry using global wood harvest input data (e.g. harvested area) is achieved by creating secondary forest stands after clear-cutting either a PNV stand or other secondary forest stands, representing cutting of primary or secondary forest, respectively. In cutting events, looping through the stands, these are cut according to stand age rules (cut oldest or youngest stands first, avoid cutting stands younger than 15 years old), allowing the allocation of wood harvest to primary forest and mature or young secondary forest. This method was used by Pugh et al. (2019) with reconstructed time series of land use from the Land Use Harmonization Project (LUH2, Hurtt et al., 2017)

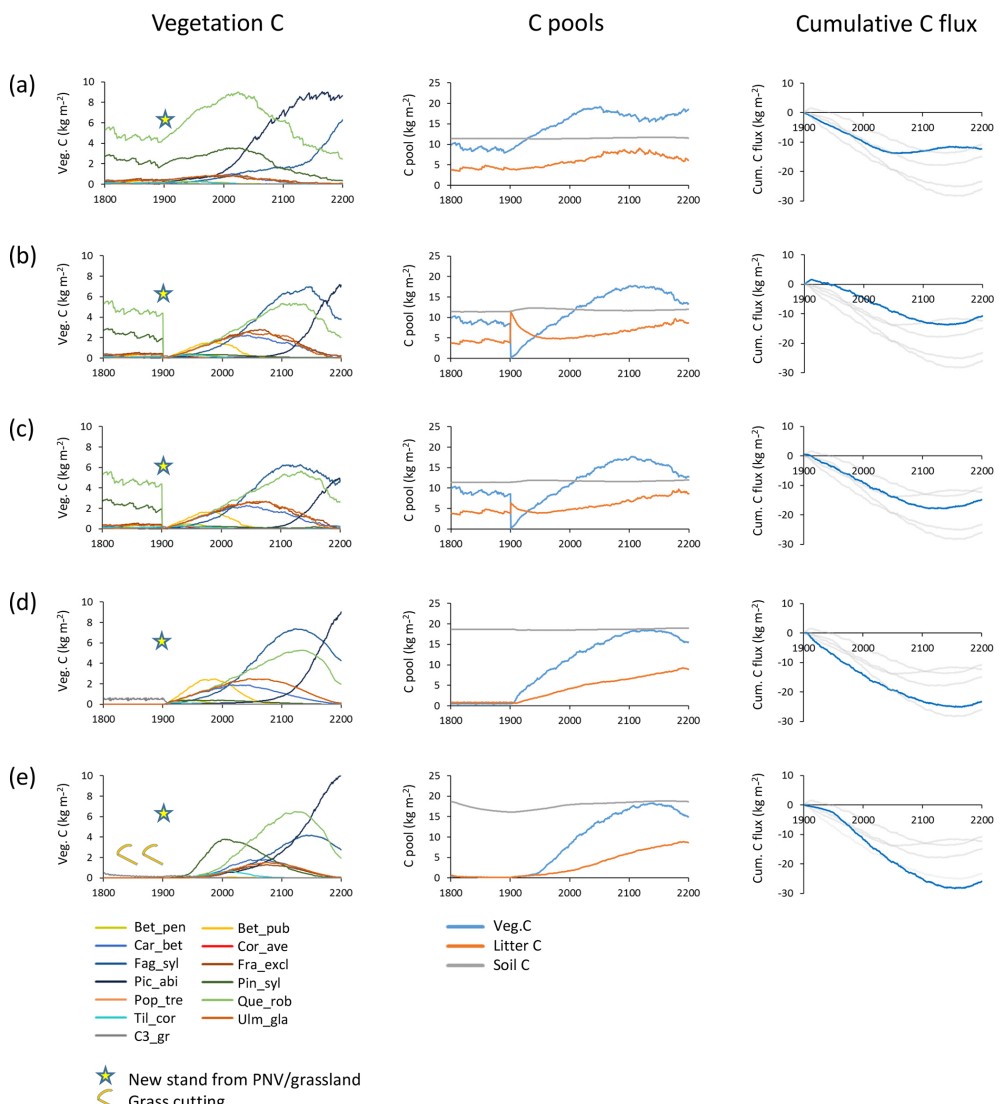

**Figure 2.** Examples of different histories and initialisations of modelled forest stands at a southern Swedish site (55.75° N, 13.75° E) with CRU-NCEP climate (recycled 1986–2015 climate after 2015). Disturbance and fire were turned off in the managed forest stands. Vegetation carbon, carbon pools (vegetation, litter and soil), and cumulative total carbon flux (negative values correspond to an uptake from the atmosphere) are shown for forest stands created in 1901 from PNV or grassland. **(a)** PNV stand with 25 patches cloned, keeping age and species structure from the spin-up period intact. **(b)** Clear-cutting of PNV stand. Harvested wood and branches left as litter. Succession from bare ground. **(c)** Clear-cutting of PNV stand. Harvested wood and part of branches removed. Succession from bare ground. **(d)** From grassland with one patch. **(e)** From intensively cut meadow with one patch, 100 % of leaves cut each year in 1800–1900. Species and PFT abbreviations are as follows: Bet_pen, *Betula pendula*; Bet_pub, *Betula pubescens*; Car_bet, *Carpinus betulus*; Cor_ave, *Corylus avellana*; Fag_syl, *Fagus sylvatica*; Fra_exc, *Fraxinus excelsior*; Pic_abi, *Picea abies*; Pin_syl, *Pinus sylvestris*; Pop_tre, *Populus tremula*; Que_rob, *Quercus robur*; Til_cor, *Tilia cordata*; Ulm_gla, *Ulmus glabra*; and C3_gr, C$_3$ grass.

### 2.3.2 Detailed forest management options

A number of forest management options can be selected at the stand type or management type level in the LPJ-GUESS instruction text file required to run a simulation and used with both even-aged and uneven-aged forestry (Table 1).

### Species selection

A forest stand may contain a full selection of tree species (as in PNV) or a selection of species defined in the management type. After a clear-cut event, or after creating a new forest stand from bare ground or grassland, selected species may be planted at defined sapling densities with or without the additional need to fall within the envelope of the bio-

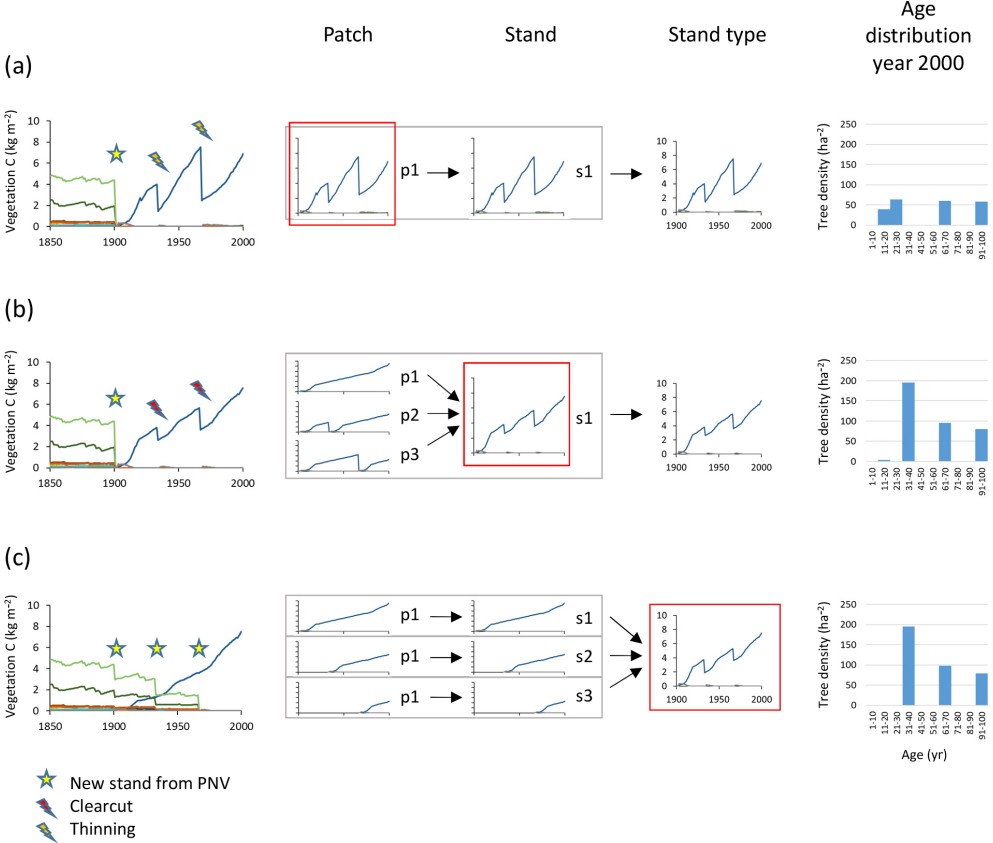

**Figure 3.** Examples of age structure setup at three different structural levels, patch, stand, and stand type. Monocultures of European beech are created from clear-cutting of PNV. The target in the year 2000 was three cohorts of 100, 67, and 33 years. **(a)** Within-patch setup with the following characteristics: one secondary stand with one patch created in 1901, thinnings in 1933 and 1967, and an age structure that depends on timing of increased light and subsequent re-establishment of seedlings. **(b)** Among-patch age setup with the following characteristics: one secondary stand with three patches created in 1901 and clear-cutting in patches 2 and 3 in 1933 and 1967 (evenly spread age distribution). **(c)** Among-stand age setup with the following characteristics: three secondary stands with one patch created in 1901, 1933, and 1967; age structure from area fraction input. Location, climate input, and species in PNV are as in Fig. 2.

climatic limits that govern PFT establishment in PNV mode (Table A1). Re-establishment can be optionally enabled or disabled for selected and unselected species. If several tree species are selected, it is possible to prescribe a target relative abundance for each species and apply cutting to regulate the mixing proportion. Relative biomass values of selected species are then monitored at 5-year intervals, and if the values deviate more than 10 %, dominant species are cut to reach the target (Fig. 4a).

## Clear-cutting

A fixed rotation period is defined at the end of which a clearcut takes place (Fig. 5a). Alternatively, a clear-cut may be triggered by attainment of a prescribed stand density limit (Fig. 5b). The timing of a number of thinning events (default 5) may be defined as fractions of the rotation period in the case of a fixed rotation period. The harvest amount (intensity) for such thinning events is defined as a fraction of cur-

rent biomass, with the option of different settings for selected and unselected species. At each thinning event, trees may be cut using alternative strategies. Available size and age criteria are (1) old or big trees first ("from above"), (2) young or small trees first ("from below"), (3) a specified harvest amount pertaining to trees above a specified diameter limit only ("threshold diameter thinning"), and (4) all sizes and ages cut equally. These may be combined with the following species criteria: (1) selected species first, (2) unselected species first, (3) separately defined harvest amounts for selected and unselected species, (4) shrubs and shade-intolerant species first, and (5) all species cut equally (Fig. 5a). In (1) and (2) size overrides age settings.

## Continuous cutting

When modelling continuous cutting, it is possible to define the same harvest parameters and cutting priority settings as described above for the clear-cutting case for two different

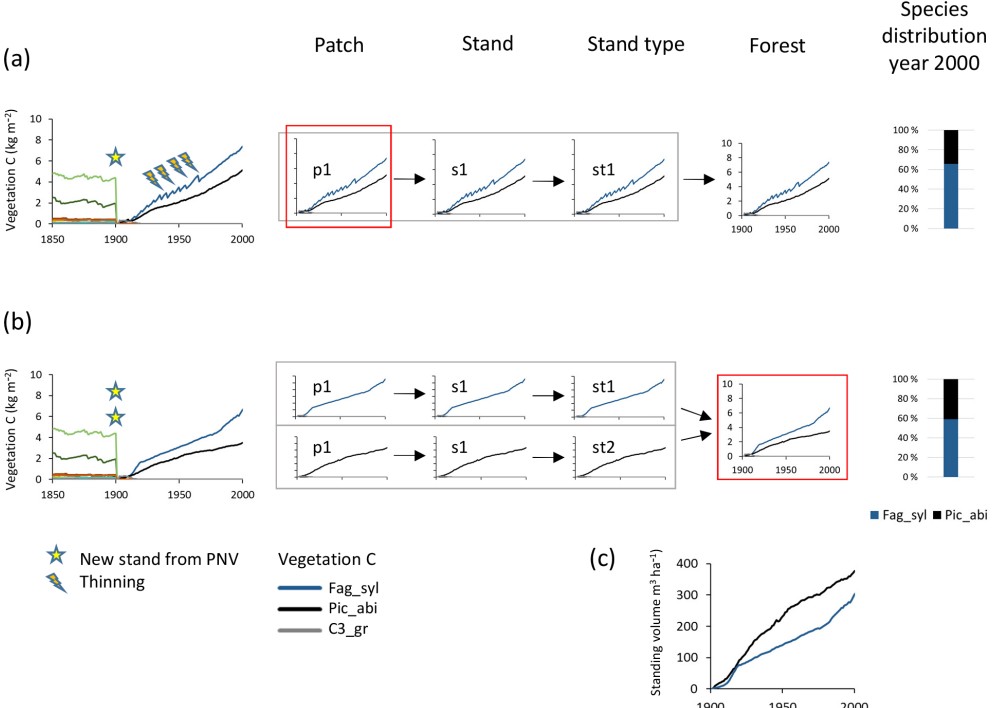

**Figure 4.** Examples of species structure setup at the patch and forest level. Beech–spruce 60 %–40 % mixes are created after clear-cutting of PNV. **(a)** Within-patch setup with the following characteristics: one secondary stand with one patch created in 1901; mixed beech–spruce with selective thinning (target cutting to a 60 %/40 % biomass ratio). **(b)** Among-stand type setup with the following characteristics: two secondary stands (beech and spruce monocultures) with one patch created in 1901 with 60 % and 40 % groundcover area fractions. **(c)** Relative development of standing volume of beech and spruce in their separate stands in **(b)**. Species abbreviations are as follows: Fag_syl, *Fagus sylvatica*; Pic_abi, *Picea abies*; C3_gr, $C_3$ grass. Location, climate input, and species in PNV are as in Fig. 2.

**Table 1.** Detailed forest management options. All management options except re-establishment can be defined in separate management types (see Fig. 1), which may be selected in a stand type rotation scheme at pre-defined calendar years.

| | | Management system | | | | |
| --- | --- | --- | --- | --- | --- | --- |
| | | Uneven-aged forestry | | Even-aged forestry | | |
| Management option | | Regeneration phase | Continuous phase | Detailed | Automated | Simplified |
| Planting | | NA | | PFT (species) selection, density | | NA |
| Re-establishment | | free or species selection or none | | | | free |
| Thinnings | preference | young or old, big or small, unselected PFTs, shrubs or shade intolerant, diameter limit | | young or old, big or small | | NA |
| | intensity | fraction of biomass | | self-thinning rule | | NA |
| | timing | fraction of rotation time | | | | NA |
| Rotation time | | length of phase | time of harvest cycle | fixed rotation | tree density | harvest demand input |
| Clear-cuts | | NA | | time | limit | stand selection rules: primary or secondary, young/old |
| Species selection cutting | | pre-defined relative species fractions | | NA | | |
| N fertilisation | | N amount evenly distributed over the whole year | | | | |
| Irrigation | | water amount required to avoid water stress in photosynthesis added to soil | | | | |
| Fire or disturbance suppression | | switch off fire and/or disturbance | | | | |
| Management change | | change management type a specific calendar year (optionally wait for clear-cut) | | | | |

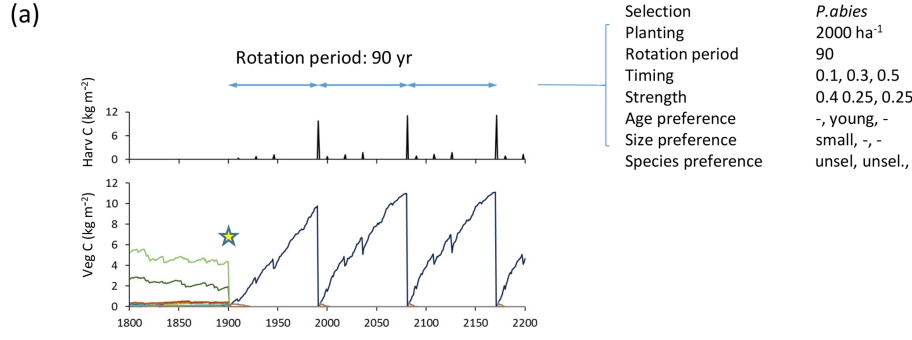

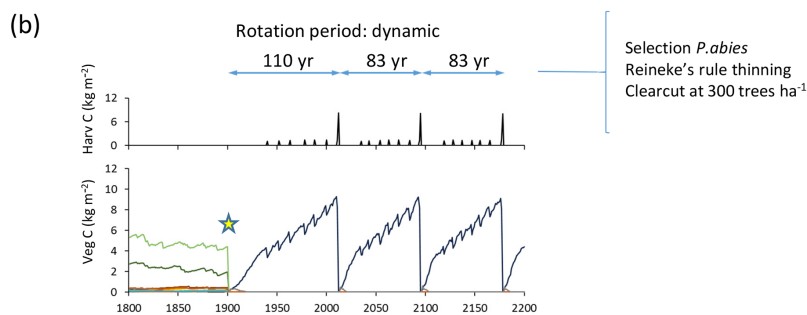

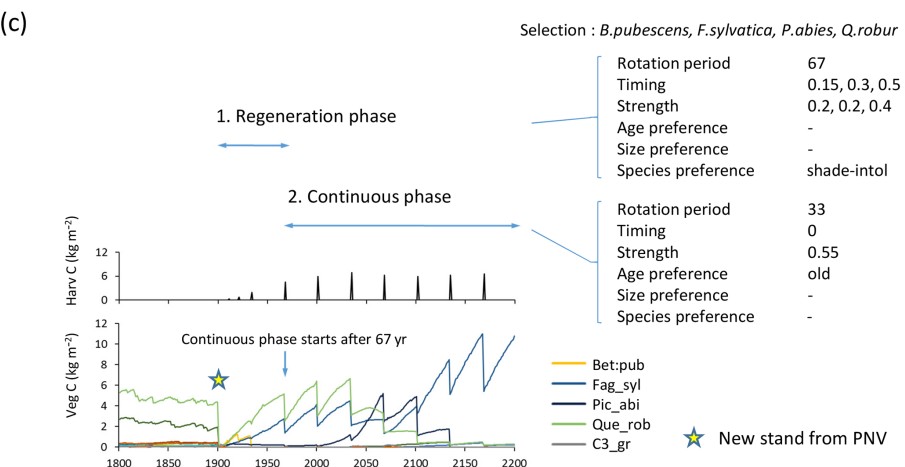

**Figure 5.** Examples of forest management settings. Forestry stands were created from clear-cutting of PNV in 1901. **(a)** Detailed clear-cut forestry: spruce monoculture with fixed rotation period and thinning parameters. **(b)** Automated clear-cut forestry: spruce monoculture with automated thinning and clear-cutting. **(c)** Continuous selection/shelterwood cutting: species selection, *B. pubescens, F. sylvatica, P. abies, Q. robur*, established after clear-cutting. Later reestablishment of all species allowed. Cutting of shade-intolerant species during a regeneration phase. Continuous partial harvest of old trees every 33 years allows establishment of young cohorts while suppressing shade-intolerant species. Species/PFT abbreviations are as follows: Bet_pen, *Betula pendula*; Fag_syl, *Fagus sylvatica*; Pic_abi, *Picea abies*; Que_ rob, *Quercus robur*; and C3_gr, C$_3$ grass. Location, climate input, and species in PNV are as in Fig. 2.

periods: the first for a specified "regeneration" time following a clear-cut and the second for a "continuous" phase in which the cutting cycle is repeated indefinitely (Fig. 5c).

**Automated wood harvest**

As an alternative to specifying thinning in clear-cut forestry in detail, a thinning scheme based on Reineke's self-thinning rule may be chosen (Fig. 5b). The implementation follows Bellassen et al. (2010):

$$\text{dens}_{\text{max}} = \frac{\alpha_{\text{st}}}{\text{Dg}^{\beta_{\text{st}}}}, \tag{1}$$

where $\text{dens}_{\text{max}}$ is stand maximum density before self-thinning (trees ha$^{-1}$), $\alpha_{\text{st}}$ and $\beta_{\text{st}}$ are fixed parameters, and Dg is the quadratic mean diameter (m),

$$\text{Dg} = \sqrt{\frac{\sum_i \text{diam}_i^2}{N}}, \tag{2}$$

where $\text{diam}_i$ is the tree diameter (m) of an individual tree and $N$ is the number of sampled trees

The parameters $\alpha_{\text{st}}$ and $\beta_{\text{st}}$ were calibrated from log–log plots of Dg and tree density, dens, from LPJ-GUESS simulations of monocultures without disturbance or re-establishment, starting from bare ground after clear-cutting of PNV (Fig. C1):

$$\log \text{Dg} = \frac{\log \alpha_{\text{st}}}{\beta_{\text{st}}} - \frac{1}{\beta_{\text{st}}} \times \log \text{dens}. \tag{3}$$

To avoid natural tree mortality occurring due to the model's self-thinning functionality, the relative density index, rdi, is monitored

$$\text{rdi} = \frac{\text{dens}}{\text{dens}_{\text{max}}} \tag{4}$$

and kept close to a target value, $\text{rdi}_{\text{target}}$, by cutting when rdi reaches $(\text{rdi}_{\text{target}} + \delta\text{rdi})$ to reach $(\text{rdi}_{\text{target}} - \delta\text{rdi})$, where

$$\delta\text{rdi} = 0.05 + \left( 0.05 \times \log\left( \frac{\text{dens}}{\text{dens}_{\text{target}}} \right) \Big/ \log\left( \frac{\text{dens}_{\text{init}}}{\text{dens}_{\text{target}}} \right) \right), \tag{5}$$

where $\text{dens}_{\text{init}}$ is the initial tree density and $\text{dens}_{\text{target}}$ is the density limit for clear-cutting (see below).

As an alternative to imposing a specified rotation length in clear-cut forestry, a clear-cut may be triggered by stand density when it is below $\text{dens}_{\text{target}}$ as in Bellassen et al. (2010).

$\text{rdi}_{\text{target}}$ and $\text{dens}_{\text{target}}$ were selected and $\alpha_{\text{st}}$ further adjusted to give rotation times around 100 years in the early 2000s in LPJ-GUESS simulations (Table A3).

### Nitrogen fertilisation and irrigation

A specified amount of plant-available nitrogen may be applied to the soil evenly distributed over the whole year (Fig. B2). With irrigation enabled, the amount of water required to avoid water stress is calculated and applied to the soil surface every year.

### Management change

To capture management changes, a new silvicultural treatment of a stand type can be prescribed any specified calendar year, changing from one specified management type to another with the next harvest event as an optional trigger (Fig. 6).

## 2.4 Demonstration simulation protocol

To demonstrate the implemented forest management functionality and its effects on simulated stand structure, composition, and productivity, we performed demonstration simulations for representative locations (grid cells) in Europe and across Europe as a whole. PNV stands were modelled using 25 replicate patches and a disturbance return time of 400 years. Managed forest stands contained only one patch except where explicitly stated (Sect. 2.5), disturbance and fire were turned off, and mortality was deterministic. In managed forest stands created after clearing the previous vegetation, this setup saves computational time and produces almost identical results compared to using multiple patches and adding the stochastic component to establishment and mortality. Parameters for European species were adopted from Hickler et al. (2012) with updated parameters (Tables A1–A2) and with the addition of *Larix decidua* (Scherstjanoi et al., 2014), *Populus tremula*, and *Ulmus glabra*.

Historic (1901–2015) monthly temperature, radiation, and precipitation data at $0.5° \times 0.5°$ resolution were taken from the station-based CRU-NCEP climate dataset (Wei et al., 2014) and atmospheric $CO_2$ concentration data from the global carbon project (Le Quéré et al., 2018). Nitrogen deposition data for 1850–2009 were taken from Lamarque et al. (2011). Simulations began with a 1300-year spin-up to initialise PNV species composition and soil and plant carbon pools. Detrended 1901–1930 climate was recycled and 1901 $CO_2$ concentration was prescribed throughout the spin-up. Nitrogen deposition data for 1850–1859 were applied before 1860 after which the historic data were used as forcing. After 2015, the 1986–2015 climate data and the 2015 $CO_2$ were recycled, and after 2009 the 2000–2009 nitrogen deposition rates were assumed.

In future climate scenario simulations, monthly temperature, radiation, and precipitation data for 1850–2100 were adopted from the IPSLCM5A-MR (Dufresne et al., 2013) GCM (global climate model) projections from the CMIP5 ensemble (Taylor et al., 2011). Projections forced by the RCP 4.5 and 8.5 future radiative forcing scenarios were used. The raw GCM climate output fields were interpolated to $0.5° \times 0.5°$ resolution and bias-corrected on a monthly basis against the CRU-NCEP 1961–1990 observational climate, following the approach of Ahlström et al. (2012). Atmospheric $CO_2$ concentration data for 1850–2100 consistent with the CMIP5 GCM forcing were used. During a 1250-year spin-up, the detrended 1850–1879 climate was recycled and the 1850 $CO_2$ and nitrogen deposition data (Lamarque et al., 2011) were used. After 2100, the 2071–2100 detrended climate data were recycled and the 2100 $CO_2$ data and the 2090–2099 nitrogen deposition data were used.

In future forest projections, either the historic environmental drivers were recycled after 2015 or future climate, $CO_2$, and nitrogen projections were used to demonstrate model behaviour under a time span of several forest rotations.

https://doi.org/10.5194/gmd-14-1-2021 Geosci. Model Dev., 14, 1–42, 2021

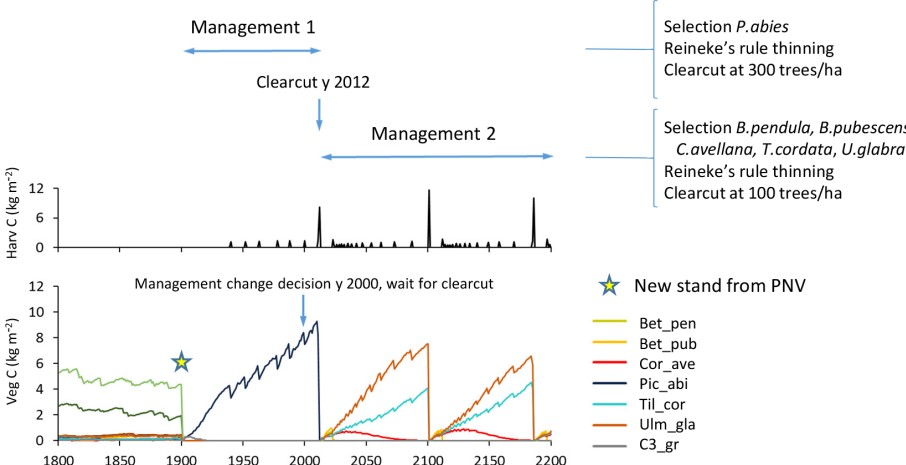

**Figure 6.** Example of management change during an ongoing simulation. Spruce monoculture changed to mixed broadleaved (both with automated thinning and clear-cutting). Management change is activated after first management has completed by a clear-cut event. Location, climate input, and species in PNV are as in Fig. 2.

## 2.5 Site-level simulations

A grid cell in southern Sweden (55.75° N, 13.75° E) was selected to demonstrate forest development under different forest stand histories and initialisation and management strategies. The setup and CRU-NCEP climate were as described in Sect. 2.4, except that three patches were used in secondary forest stands when illustrating among-patch age structure setup.

Four datasets of European beech and Norway spruce monoculture stand time series (1–21 points in time) of standing volume and harvested volume were used in simulations to initialise species and age structure, assuming a landscape distribution of even-aged stands. The stands were located in central and southern Germany (GER-Bav, GER-C, GER-CS) and northern Slovenia (SLO, beech only) (Appendix D, Table D1). The model setup and input climate data were as described in Sect. 2.4. Three different harvest strategies were used: no harvest, detailed harvest from observations, and automated thinning and clear-cutting (Sect. 2.3.2). The setup of the detailed harvest for stands from the different datasets differed slightly depending on the number of harvest data points. For the stands from the GER-Bav, GER-C, and SLO data sources (3–21 data points per stand), the harvest data (fraction of biomass) were used in the simulations at the reported timings. During the time period prior to the first harvest data point, mean harvest intensities from the harvest data were used, in the case of GER-Bav and GER-C converted to fit a 5-year harvest interval, while in the case of SLO keeping the 10-year interval used in the sampling. The GER-CS data contain only one harvest data point for the whole stand lifetime (100 years). In this case, harvests were performed at 5-year intervals during the whole simulation using the calibrated harvest intensity values required to obtain a cumula-

tive harvest fraction equal to the reported harvest fraction for the whole 100-year period. Thinnings in the detailed harvest simulations were performed equally for the different cohorts to obtain some regeneration of saplings in old stands. The automated thinning and clear-cutting method used the parameter settings in Table A3 and thinnings from below started at a stand age of 10 years.

## 2.6 European simulations

### 2.6.1 Forcing data

To constrain European secondary forest age and species structure in the model to the actual state of the forests, we used the global forest age dataset GFAD (Poulter et al., 2019; Pugh et al., 2019), describing the 0.5° × 0.5° grid cell fraction coverage of 14 total 10-year cohorts of the forest types needleleaf evergreen (NE), needleleaf deciduous (ND), broadleaf evergreen (BE), and broadleaf deciduous (BD) in the year 2010. For Europe, the data were based on The European Forest Information SCENario Model (EFIS-CEN, European Forest Institute (EFI)) in the 2000s. European forests (excluding Russia outside of the Kaliningrad region, Georgia, Iceland, and Cyprus in this study) consisted of 0.6 million km$^2$ old-growth forests (>140 years, denoted as "old-growth" forest henceforth, not implying pristine forests) and 1.8 million km$^2$ regrowth forests in 2010 according to GFAD, together making up about 43% of the European land area. This is higher than other estimates (e.g. Forest Europe, 2015, 35%) and is a result of the construction of the GFAD database from MODIS 5.0, with the inclusion of shrubland. In GFAD, regrowth forests are the result of both natural disturbances and human interventions, but since only 0.7% of European forests are pristine (Sabatini et al., 2018), the whole regrowth forest area was assigned to sec-

ondary forest in this study. The oldest forest class in GFAD (>140 years) contains artefacts manifested as, e.g. BE occurrences in northern Europe, so the forest type information in this part of the dataset was not used.

The EFI Tree species map describes the spatial distribution (fraction of land area) of 20 tree species groups at $1 \times 1$ km resolution (Brus et al., 2011). The map is based on ICP-Forest Level I plot data combined with National Forest Inventory (NFI) data of 18 countries. In areas with NFI data, spatial interpolation of the plot data was used, whereas in areas without NFI data, statistical relationships between tree species and covariates (soil, biogeography, and bioindicators) were used (Brus et al., 2011). The EFI Tree species map was aggregated to $0.5° \times 0.5°$ resolution in this study and was used to further refine the species distribution derived from GFAD.

The structure of European forests in 2010 was reconstructed by using a combination of the GFAD age database and the EFI Tree species map. For each grid cell, the most common species or species group within the GFAD NE and BD forest types was obtained from the EFI Tree species map and these were then mapped to LPJ-GUESS tree species or species groups (Table C1, Fig. C2). In the multi-species LPJ-GUESS groups, species compete with each other for resources (see Sect. 2.1). BE was mapped to *Quercus ilex* and ND to *Larix decidua*, the only available PFTs in the model to represent these two functional tree classes.

### 2.6.2 Modelling current and future European managed forests

Secondary forest stands were created in the model from 1871 to 2010 to obtain the GFAD age (1–140 years) distribution in 2010, and species selections were planted (without climate restrictions for NE and ND stands to bypass establishment temperature limits used in PNV). The oldest forest class in GFAD (>140 years) was modelled as PNV and was not subject to any management (see Sect. 2.6.1). In secondary stands, automated thinning and clear-cutting (see Sect. 2.3.2) were implemented using the parameters in Table A3. Thinnings from below started at a stand age of 10 years, and clear-cutting started after the year 2010. Clear-cuts of stands that passed below the tree density limit before 2011 were distributed over the years 2011–2020. In an alternative simulation with identical stand structure setup, thinning and clear-cutting were turned off.

To perform a limited sensitivity test of some of the uncertainties in land-use and residue removal assumptions, additional alternative simulations were performed: a simulation where a fraction (as in standard harvest) of the biomass of trees killed in natural disturbance events in old-growth forests was removed from year 1871, simulating an extensive wood harvest scheme and two simulations where the leaf removal fraction in harvest events was set to 10 % and 0 %, respectively, instead of the standard 30 % value.

### 2.6.3 Calculation of output variables

Growing stock, net annual increment (NAI), and harvested volume were calculated from vegetation carbon, net ecosystem exchange (NEE), and total carbon of harvested trees, respectively, by multiplying with expansion factors for each country, ranging from 1.1 to 3.5 (mean 2.7) $m^3 t C^{-1}$, derived from vegetation carbon and growing stock volumes reported by Forest Europe (2015). Carbon sink ($= -$NEE) is defined as the difference in the sum of vegetation and soil carbon pools between 2 consecutive years plus the removed harvested carbon, not taking into account the fate of wood products and residues following removal from the site. Similarly, NAI is defined as the difference in growing stock volume between 2 consecutive years plus the removed harvested volume. Harvested carbon is not included in the total carbon pool and includes both wood products and removed wood residues. The forested area in 2010 as defined by GFAD and Forest Europe (2015) was 2.4 and 2.0 million $km^2$, respectively, excluding Georgia, Iceland, Cyprus, Malta, and Russia but including the Kaliningrad region and the European part of Turkey. The forest area available for wood supply (FAWS), for GFAD defined as the secondary forest area in 2010 was 1.8 and 1.6 million $km^2$ for GFAD and Forest Europe (2015), respectively.

## 3 Results

### 3.1 Implications of secondary forest initialisation and land-use history

Secondary forest stand initialisation and land-use history have long-term effects on the development of tree species distribution, productivity, and carbon fluxes in the model (Fig. 2). When the age distribution and species composition from spin-up is retained in each patch (i.e. cloning PNV), both the warming climate in the 20th century and the prevention of fires and other disturbances result in an increase in tree biomass and a tree species shift from a *Q.robur*–*P.sylvestris*-dominated forest landscape to a forest increasingly dominated by the shade-tolerant species *P.abies* and *F.sylvatica* in an example forest simulated at a southern Swedish site (Fig. 2a). Older patches contribute to an early stagnation of the carbon sink. A forest stand created after clear-cutting PNV displays a mixed broadleaf forest with a late establishment and dominance by *P.abies* (Fig. 2b and c). Leaving harvested biomass on site results in an extended litter-induced carbon source (Fig. 2b). When the previous land-use history is grassland, the initial dominance by shade-intolerant species is more pronounced and the slow accumulation of the litter pool results in a stronger and more persistent carbon sink (Fig. 2d, e). Soil carbon and nitrogen depletion due to intensive harvest of the previous grassland influences productivity, succession of tree species, and carbon

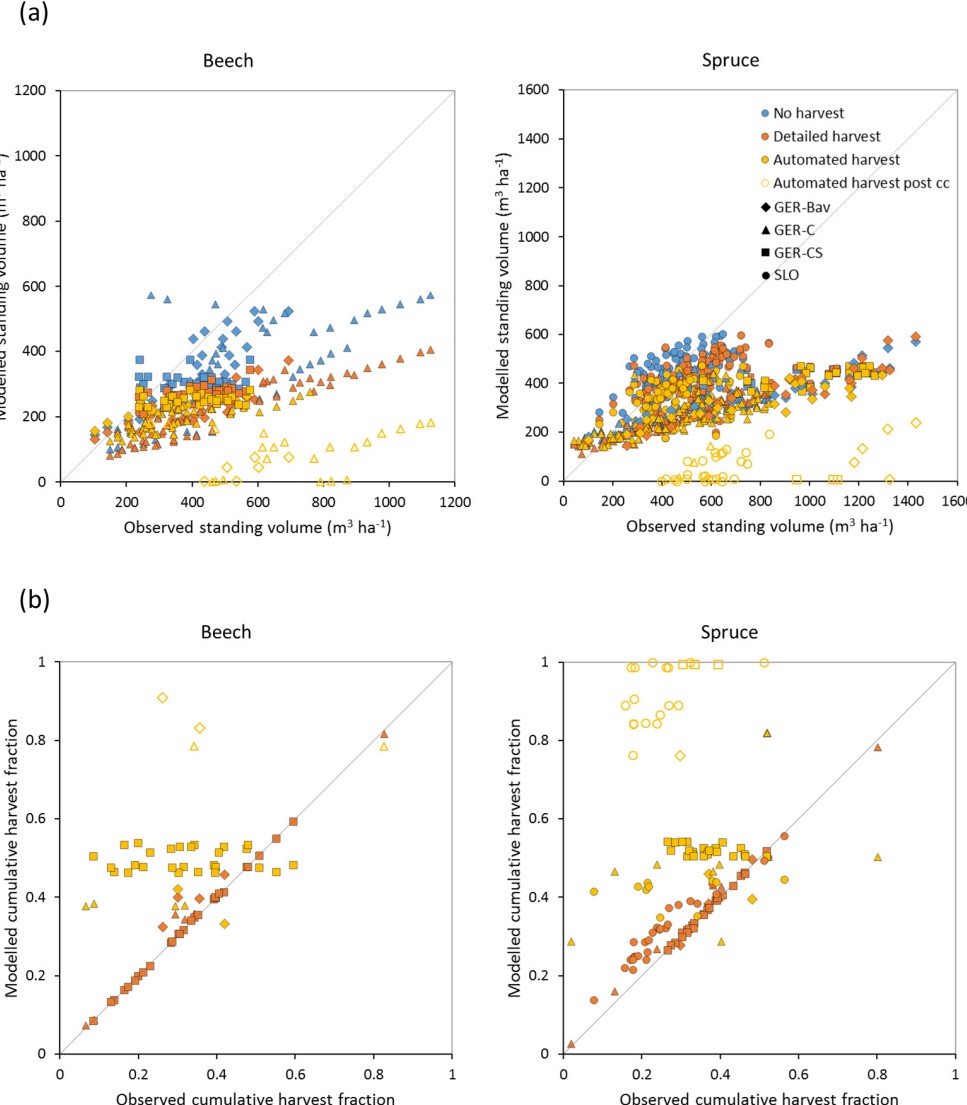

**Figure 7.** Modelled and observed standing volume **(a)** and cumulative harvest fraction during the measurement period **(b)** for European beech and Norway spruce stands in Germany (Bavaria: GER-Bav; central Germany: GER-C; central and southern Germany: GER-CS) and Slovenia (SLO). Simulations were performed without harvest, with detailed harvest, and with automated thinning and clear-cutting. Data points from the automated harvest simulation after clear-cutting occurred are plotted with unfilled symbols (automated harvest post cc).

sink capacity of the secondary forest: initial tree growth is delayed by several decades, the dominant shade-intolerant species is *P.sylvestris* rather than *B.pubescens*, and *Q.robur* competes more successfully than under normal soil nitrogen (Fig. 2e). In addition, the long-term carbon sink is larger than in any other option. The notable differences in tree species succession and the timing and magnitude of the carbon sink using the different stand creation options illustrate the importance of land-use history for modelling secondary forest stands.

## 3.2 Choosing between different model age and species structure and harvest alternatives

The choice between the different age and species structure setup options depends on whether competition between species and cohorts within patches is required or not (Figs. 3–4). Also, the desired level of detail of the age structure might decide whether to use a simplified setup or a detailed structure with many separate stands, increasing computation time. Setups using separate stands for each species–age combination offer the possibility of reflecting regional distributions based on inventory data but will not represent competition correctly e.g. in mixed forests.

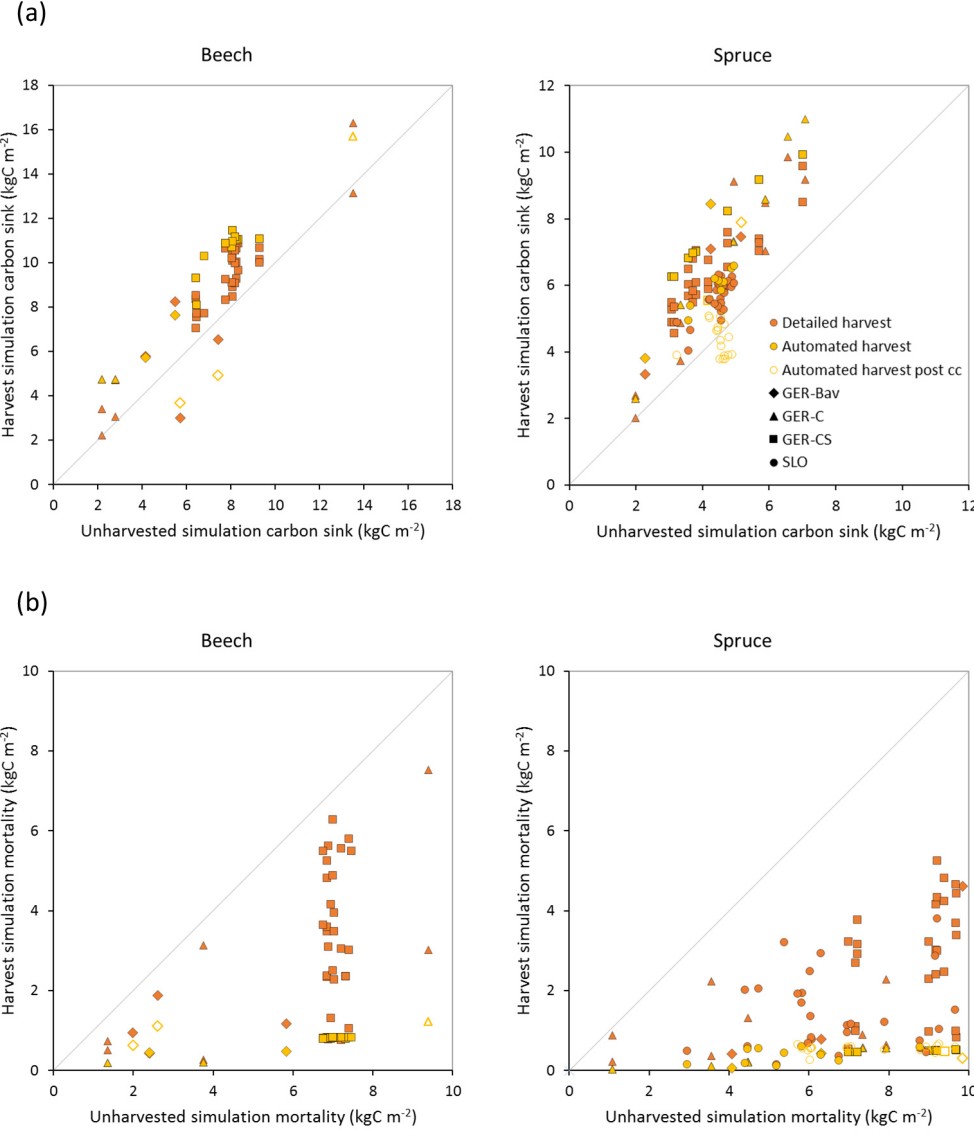

**Figure 8.** Modelled carbon sink **(a)** and cumulative mortality **(b)** for the same time periods as in Fig. 7b in simulations with detailed and automated harvest compared to a simulation without harvest of European beech and Norway spruce stands in Germany (Bavaria: GER-Bav; central Germany: GER-C; central and southern Germany: GER-CS) and Slovenia (SLO). Data points from the automated harvest simulation after clear-cutting occurred are plotted with unfilled symbols (automated harvest post cc).

Although management changes during the course of a simulation may be prescribed, using detailed but static harvest methods would not reflect foresters' choice of gradual adaptation of harvest parameters under changing $CO_2$ and climate conditions in future scenarios. In these cases, the simplified dynamic harvest methods might be a better option (Fig. 5b).

### 3.3 Central European site simulations of managed forest

Central European beech and Norway spruce stands were modelled with three harvest alternatives: no harvest, detailed harvest based on reported harvested volumes, and automated

thinning and clear-cutting. The model was not able to reach the high productivity of beech and spruce stands in Germany. The modelled standing volumes of these stands were relatively accurate at low standing volumes but about 2–3 times underestimated at high observed standing volumes (Fig. 7a). The correlation between modelled and observed German standing volume was generally good: $r^2 = 0.64$ and 0.86 for pooled detailed harvested beech and spruce stands, respectively, and $r^2 = 0.51$ and 0.79 for the corresponding unharvested stands. The Slovenian spruce standing volume levels were better represented by the model, but the correlation with observations was weaker ($r^2 = 0.36$ for detailed harvested stands and 0.21 for unharvested stands). The addi-

**Table 2.** Modelled and observed forest vegetation carbon stock in Europe.

| | LPJ-GUESS (this study)[1] | Liu et al. (2015)[2] | Pan et al. (2011) | Forest Europe |
|---|---|---|---|---|
| **Veg C (Pg C)** | | | | |
| **Europe[3]** | | | | |
| 2000 | 13.8 (14.3) | 11.1 TS1 | 11.8 | 10.2 |
| 2007 | 14.1 (14.7) | 11.6 | 13 | |
| 2010 | 14.3 (15.0) | | | 11.8 |
| 2015 | 14.2 (15.8) | | | 12.5 |
| **EU-28 + Switzerland[4]** | | | | |
| 2000 | 11.3 (11.7) | | | 8.3 |
| 2010 | 11.6 (12.2) | | | 9.4 |
| 2015 | 11.4 (12.9) | | | 10.0 |
| **Veg C (kg C m$^{-2}$)** | | | | |
| **Europe[3]** | | | | |
| 2000 | 5.5 (5.7) | 5.5 | 5.9 | 5.3 |
| 2007 | 5.7 (5.9) | 5.7 | 6.4 | |
| 2010 | 5.7 (6.0) | | | 5.9 |
| 2015 | 5.7 (6.4) | | | 6.3 |
| **EU-28 + Switzerland[4]** | | | | |
| 2000 | 5.8 (6.0) | | | 5.3 |
| 2010 | 5.9 (6.2) | | | 5.9 |
| 2015 | 5.9 (6.6) | | | 6.2 |

[1] Values in parentheses are for a simulation without wood harvest in secondary forest. [2] AG biomass is 79 % of total biomass.
[3] Excluding Georgia, Iceland, Cyprus, Malta, and Russia but including the Kaliningrad region and the European part of Turkey in LPJ-GUESS data. [4] Cyprus and Malta are excluded.

tion of thinning in the simulations produced the largest difference in standing volume in some of the beech stands while the spruce stands were less affected. The modelled cumulative harvest intensities in the detailed harvest alternative were close to or slightly higher (due to thinning before the period with harvest data) than reported harvest intensities (Fig. 7b). Although the cumulative harvest in the automated harvest alternative was almost always more extensive over the modelled stand lifetime compared to the detailed harvest alternative (Fig. 7b), the standing volume was only moderately affected (Fig. 7a). The automated harvest standing volume correlations with observations were, as expected, weaker than for the detailed harvest simulations: $r^2 = 0.39$ and 0.76 for German beech and spruce stands, respectively, and 0.17 for Slovenian spruce stands. Both harvest alternatives increased the carbon sink at most sites and reduced mortality at all sites compared to a simulation without harvest (Fig. 8). The automated harvest led to very low levels of mortality.

### 3.4 Europe-wide simulations of managed forest

Dominant tree species in managed forests based on the EFI species map differ from PNV simulations in large parts of Europe. In central and eastern Europe, broadleaved species are to a large degree replaced by needleleaved species in managed forests, especially by *P. sylvestris*, but since old-growth forest is modelled as PNV in this study because of artefacts in the >140-year data (see Sect. 2.6.1), the dominance by needleleaves in this region seen in the original EFI data is moderated in the total forest landscape (Figs. C3, C4).

For the European continent, the modelled mean vegetation carbon density (5.7 kg C m$^{-2}$) and growing stock (156 m$^3$ ha$^{-1}$) in 2010 and NAI (5.4 m$^3$ ha$^{-1}$ yr$^{-1}$) in 2001–2010 in a simulation with thinning is close to observations (Tables 2, 4). The total carbon pool (24.2–24.3 kg C m$^{-2}$) and soil plus litter pool in 2000–2010 (18.5–18.6 kg C m$^{-2}$) is 21 %–64 % and 34 %–80 % higher than reported values, respectively, while NEE in 2000–2007 (ca. $-0.08$ kg C m$^{-2}$ yr$^{-1}$) is a sink 63 % the size of reported values (Table 3). Fellings including clear-cuts of old-growth forests and thinnings in regrowth forests (5.0 m$^3$ ha$^{-1}$ yr$^1$) and thinnings in regrowth forests only (3.0 m$^3$ ha$^{-1}$ yr$^{-1}$) in 2001–2010 are comparable to observed fellings (3.4 TS2 m$^3$ ha$^{-1}$ yr$^{-1}$) (Table 4). Simulated results for the EU-28 + Switzerland countries were closer to reported values than for the whole of Europe for most of the above variables (Tables 2–4).

**Table 3.** Modelled and observed total carbon stock, soil plus litter carbon, and net ecosystem exchange (NEE) in European forests.

| | LPJ-GUESS[1] | Pan et al. (2011)[2] | Forest Europe[3] |
|---|---|---|---|
| **Total C stock (Pg C)** | | | |
| Europe | | | |
| 2000 | 60.3 (62.3) | 39.3 | |
| 2007 | 60.4 (62.8) | 40.9 | |
| 2010[c] | 60.5 (63.1) | | 29.3 |
| EU-28 + Switzerland | | | |
| 2010[c] | 48.6 (50.7) | | 25.5 |
| **Total C stock (kg C m$^{-2}$)** | | | |
| Europe | | | |
| 2000 | 24.2 (25.0) | 19.7 | |
| 2007 | 24.2 (25.2) | 20.0 | |
| 2010[c] | 24.3 (25.6) | | 14.8 |
| EU-28 + Switzerland | | | |
| 2010[c] | 24.9 (26.1) | | 15.9 |
| **Soil + Litter C stock (Pg C)** | | | |
| Europe | | | |
| 2000 | 46.5 (48.0) | 27.6 | |
| 2007 | 46.3 (48.1) | 28.0 | |
| 2010[c] | 46.2 (48.2) | | 17.5 |
| EU-28 + Switzerland | | | |
| 2010[c] | 37.0 (38.6) | | 16.1 |
| **Soil + Litter C (kg C m$^{-2}$)** | | | |
| Europe | | | |
| 2000 | 18.6 (19.2) | 13.9 | |
| 2007 | 18.5 (19.3) | 13.7 | |
| 2010[c] | 18.5 (19.3) | | 10.3 |
| EU-28 + Switzerland | | | |
| 2010[c] | 19.0 (19.8) | | 10.8 |
| **NEE (Pg C yr$^{-1}$)** | | | |
| Europe | | | |
| 1990–1999 | −0.188 (−0.141) | −0.30 | |
| 2000–2007 | −0.212 (−0.153) | −0.27 | |
| **NEE (kg C m$^{-2}$ yr$^{-1}$)** | | | |
| Europe | | | |
| 1990–1999 | −0.075 (−0.056) | −0.154 | |
| 2000–2007 | −0.085 (−0.061) | −0.134 | |

[1] Values in parentheses are for a simulation without wood harvest in regrowth forest. [2] Litter includes dead wood.
[3] Forest Europe soil and litter carbon data missing for Bosnia, Croatia, Greece, Hungary, Macedonia, Moldova, Montenegro, Norway, and Portugal. Forest Europe total carbon and soil plus litter carbon data for 2000 and 2015 are excluded due to fewer countries with data. Europe area definition is as in Table 2.

Modelled vegetation carbon, total carbon pool, growing stock, NAI, and fellings for individual European countries show varying levels of agreement with reported values, with the best fit for vegetation carbon and growing stock ($r^2 = 0.49$ and 0.72, respectively) and the least for NAI ($r^2 = 0.06$) (Figs. 9–11, E1–E5). Modelled mean European total thinning fractions of produced wood over the whole rotation period in stands clear-cut in 2011–2020 were 0.4 for BD and 0.5 for NE (not shown). Total thinning fractions of NAI for individual countries in 2001–2010 were between 0.35 and 0.6,

with a total European mean of 0.53 (Figs. E4–E5). The corresponding annual thinning intensities of growing stocks were 0.8 % to 3.3 %, with a mean of 1.9 % (Figs. E3, E5).

Carbon pools and fluxes were partitioned into old-growth and regrowth forest components (modelled as PNV and secondary forest stands, respectively) (Fig. 12, Tables 5–6). Modelled European old-growth and regrowth forests have about equally sized vegetation carbon pools in 2000 (about 7 Pg C each) but with a downward trend for old-growth forests in 2001–2010 driven by a reduction in area. The

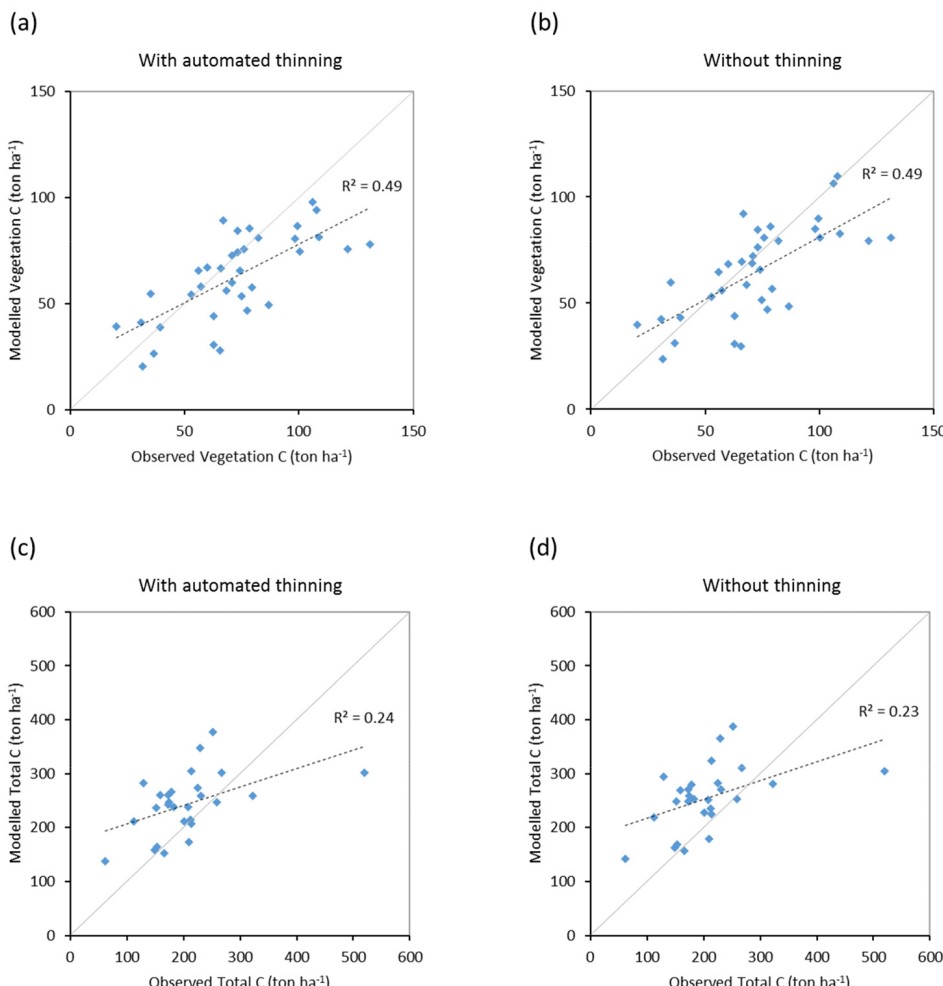

**Figure 9.** Modelled and observed (Forest Europe, 2015) values for individual European countries, excluding Georgia, Iceland, Cyprus, Malta, and Russia, in 2010. Vegetation carbon **(a)** simulation with automated thinning and **(b)** simulation without thinning. Total carbon pool **(c)** simulation with automated thinning and **(d)** Simulation without thinning. In **(c)** and **(d)**, countries missing Forest Europe soil data, i.e. Bosnia, Croatia, Greece, Hungary, Macedonia, Moldova, Montenegro, Norway, and Portugal, are also excluded.

vegetation carbon density in old-growth forests, increasing from 8.5 to $9.2\,\mathrm{kg\,C\,m^{-2}}$ between 2000 and 2015, is about twice the value in regrowth forests, increasing from 4.0 to $4.5\,\mathrm{kg\,C\,m^{-2}}$ between 2000 and 2015. This vegetation carbon difference is reflected in the difference between old-growth and regrowth forest total carbon pool density (ca. 27 and $23\,\mathrm{kg\,C\,m^{-2}}$, respectively), while the soil plus litter carbon is slightly higher (1.5 %) in regrowth forests (Table 5). The modelled forest carbon sink $(= -\mathrm{NEE})$ (2001–2010: $0.23\,\mathrm{Pg\,C\,yr^{-1}}$) is dominated by regrowth forests ($0.20\,\mathrm{Pg\,C\,yr^{-1}}$ or $0.12\,\mathrm{kg\,C\,m^{-2}\,yr^{-1}}$), compared to $0.03\,\mathrm{Pg\,C\,yr^{-1}}$ or $0.04\,\mathrm{kg\,C\,m^{-2}\,yr^{-1}}$ in old-growth forests (Table 6).

For the European continent, including thinning in the simulation reduced total forest vegetation carbon, soil plus litter carbon, total carbon pool, and growing stock in 2010 by 3 %–5 %; increased the magnitude of NEE in 2000–2007 by 39 %; and increased NAI in 2001–2010 by 100 % compared to a simulation without thinning (Figs. 13–14, Tables 2–4). In regrowth forests, including thinning reduced vegetation carbon by 6 %–7 %, soil plus litter carbon, and the total carbon pool by 5 %–6 % in 2000–2010 and increased the magnitude of NEE in 1991–2010 by 41 % (Tables 5–6). The average thinning rate on regrowth forest land was 1.9 % of wood biomass per year in 2001–2010. Including thinning generally improved the match of simulations with observed data. The increased regrowth forest carbon sink seen in a simulation with thinning ($0.12\,\mathrm{kg\,C\,m^{-2}\,yr^{-1}}$) (Fig. 12) is associated with a strong reduction of natural mortality ($-80\,\%$ in 1991–2015) in regrowth forest stands, induced by thinning and, after 2010, rejuvenation of regrowth forest stands resulting from clear-cutting (Fig. E6). The reduced natural mortality following thinning results in a lower soil respiration (Fig. E7).

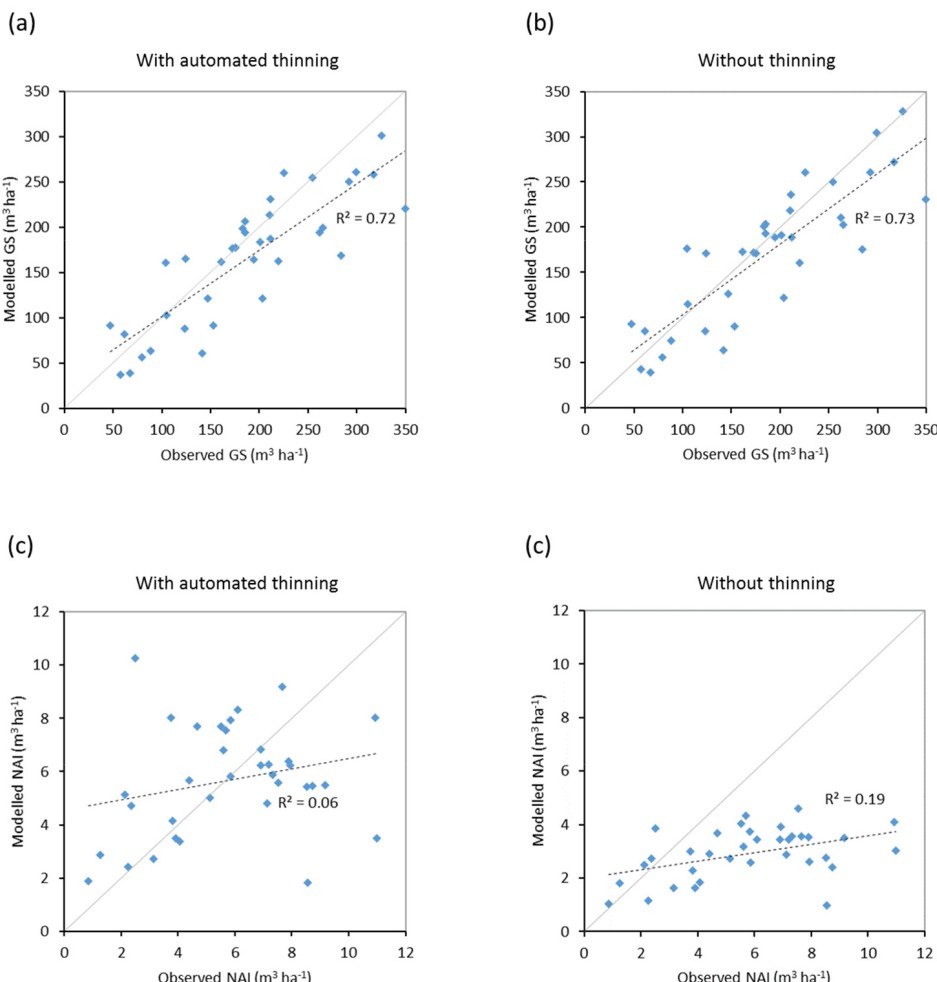

**Figure 10.** Modelled and observed (Forest Europe, 2015) values for individual European countries. Growing stock (GS) in 2010 **(a)** simulation with automated thinning and **(b)** simulation without thinning. Net annual increment (NAI) in 2001–2010 **(c)** simulation with automated thinning and **(d)** simulation without thinning. Included countries are as in Fig. 9a.

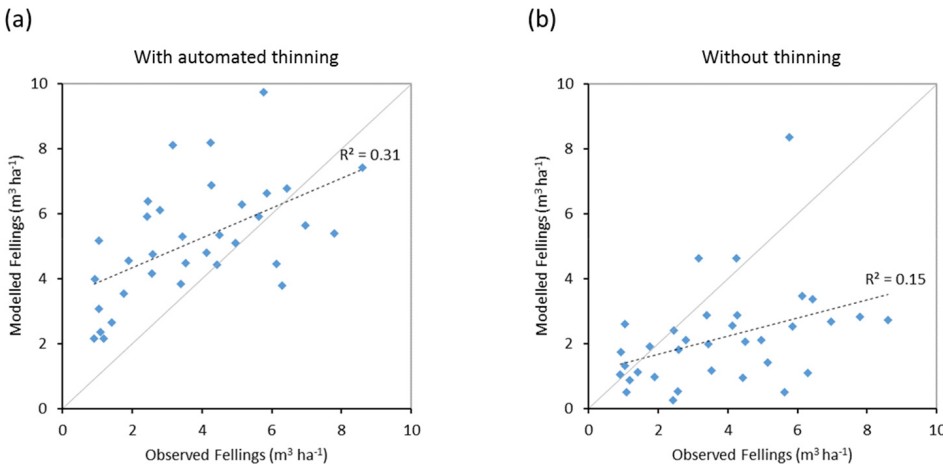

**Figure 11.** Modelled and observed (Forest Europe, 2015) yearly fellings for individual European countries in 2001–2010 showing a **(a)** simulation with automated thinning and a **(b)** simulation without thinning (clear-cutting at creation of secondary forest). Included countries are as in Fig. 9a.

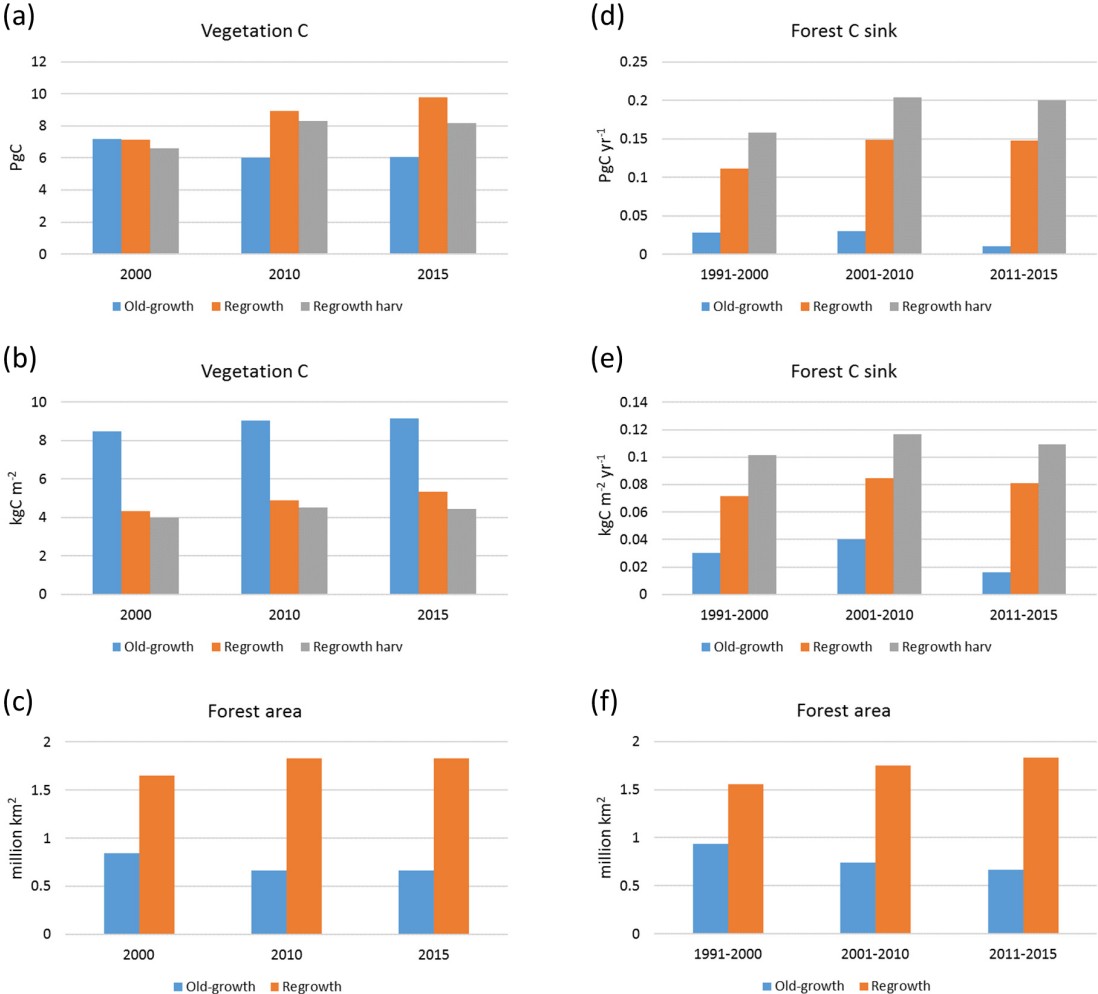

**Figure 12.** Modelled European forest vegetation carbon for 2000, 2010, and 2015 and carbon sink ($= -$NEE) for the periods 1991–2000, 2001–2010, and 2011–2015, separated into old-growth and regrowth forests (with and without wood harvest in regrowth forest): **(a)** vegetation carbon; **(b)** vegetation carbon per area; **(c)** old-growth and regrowth forest area in 2000, 2010, and 2015; **(d)** total forest carbon sink; **(e)** mean forest carbon sink per area; **(f)** old-growth and regrowth forest area in 1991–2000, 2001–2010, and 2011–2015.

In a simulation with removal of biomass during disturbance events in the old-growth stands (not shown), the carbon sink in this forest class increased to $0.04\,\mathrm{Pg\,C\,yr^{-1}}$ or $0.05\,\mathrm{kg\,C\,m^{-2}\,yr^{-1}}$ in 2001–2010 compared to a standard simulation, increasing the total forest carbon sink in the same period by 7 % to $0.25\,\mathrm{Pg\,C\,yr^{-1}}$. Soil plus litter carbon in the old-growth forest was reduced by 2.4 % in 2010 and by 0.7 % in the regrowth forest, reducing the total soil plus litter pool by 1.3 %. Total vegetation carbon increased by 0.24 % in 2010.

Simulations with alternative settings of leaf removal fractions during harvests of 10 % or 0 %, instead of 30 % in the standard simulation (not shown), decreased the total carbon sink in 2001–2010 by 0.9 % and 1.3 %, respectively, resulting from an increased soil respiration of 0.3 % and 0.4 %, respectively, partially offset by an increase in NPP by 0.06 % and 0.09 %, respectively. Vegetation carbon increased by 0.08 %

and 0.13 % and soil plus litter carbon increased by 0.07 % and 0.10 % in these simulations.

## 3.5 Robustness of automated harvest methods under future climates

To demonstrate the automated harvest methods in which thinning intensity and rotation times are adjusted to maintain standing stock when stand productivity changes in response to forcing conditions, we used $CO_2$ and climate projections in extended simulations with an otherwise identical setup as in the Europe-wide historic simulations. A significant modelled increase in NAI is accompanied by shorter rotation periods (Fig. E8), while a stable vegetation pool in managed forest is maintained (Fig. E9). The mean thinning fraction of the total harvest over the rotation for NE and BD stands increased over the 21st and 22nd centuries from 0.50

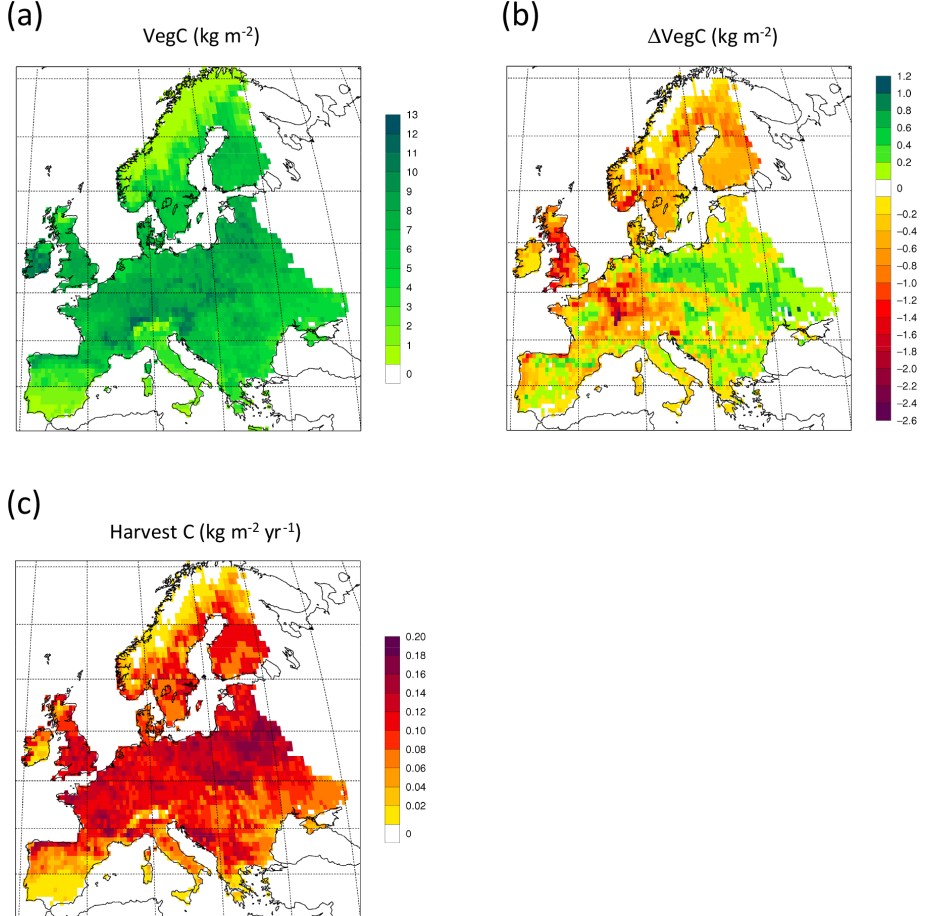

**Figure 13.** Simulated forest **(a)** vegetation carbon 2010 in a simulation with thinning; **(b)** vegetation carbon 2010 difference between simulations with and without wood harvest in regrowth forest. **(c)** Mean 2001–2010 harvested carbon during thinning in regrowth forest.

to 0.53 and 0.40 to 0.46, respectively, for both the RCP4.5 and RCP8.5 simulations (not shown).

## 4 Discussion

LPJ-GUESS representations of unmanaged forest have previously been shown to compare favourably with observed forest vegetation succession, growth, stand structure, biomass, and regrowth timescales (Smith et al., 2001, 2014; Pugh et al., 2019), and land-use and land-cover change (LULCC) functionality has been included in the model since version 4.0 (Lindeskog et al., 2013). In a recent global study that used the model to analyse the carbon stocks of old-growth and regrowth forests (modelled as primary and secondary forest stands, respectively), without applying wood harvest (Pugh et al., 2019), the total forest carbon sink was found to be about 50 % of values reported by Pan et al. (2011) based on upscaled inventory data. Disregarding wood harvest has been identified as causing underestimation of carbon sinks in vegetation models (Zaehle et al., 2006; Ciais et al., 2008). In an effort to improve the ability to simulate car-

bon pools and fluxes on managed land, we introduced new forest management options into LPJ-GUESS v4.0 and provide a comprehensive description of forest initialisation and wood harvest alternatives. The initialisation and harvest alternatives in the model are tailored to enable available forest inventory data and harvest information to be used to initialise and guide simulations. Ideally, both age and species structure, as well as land-use history and current wood harvest strategy, should be taken into account, but this is not always possible for simulations with a large spatial extent because of limited data availability. To demonstrate a possible workaround, we used an automated thinning and clearcutting alternative to represent European regrowth forests, initialised on the basis of inventory-based age and species data but without wood harvest or LULCC data input. In simulations of central European beech and spruce stands, the automated thinning method was shown to result in similar modelled standing volume but often in a higher carbon sink compared to a more detailed harvest scheme based on reported harvest intensities (Fig. 7). The harvested volume was generally substantially higher in the automated thinning sim-

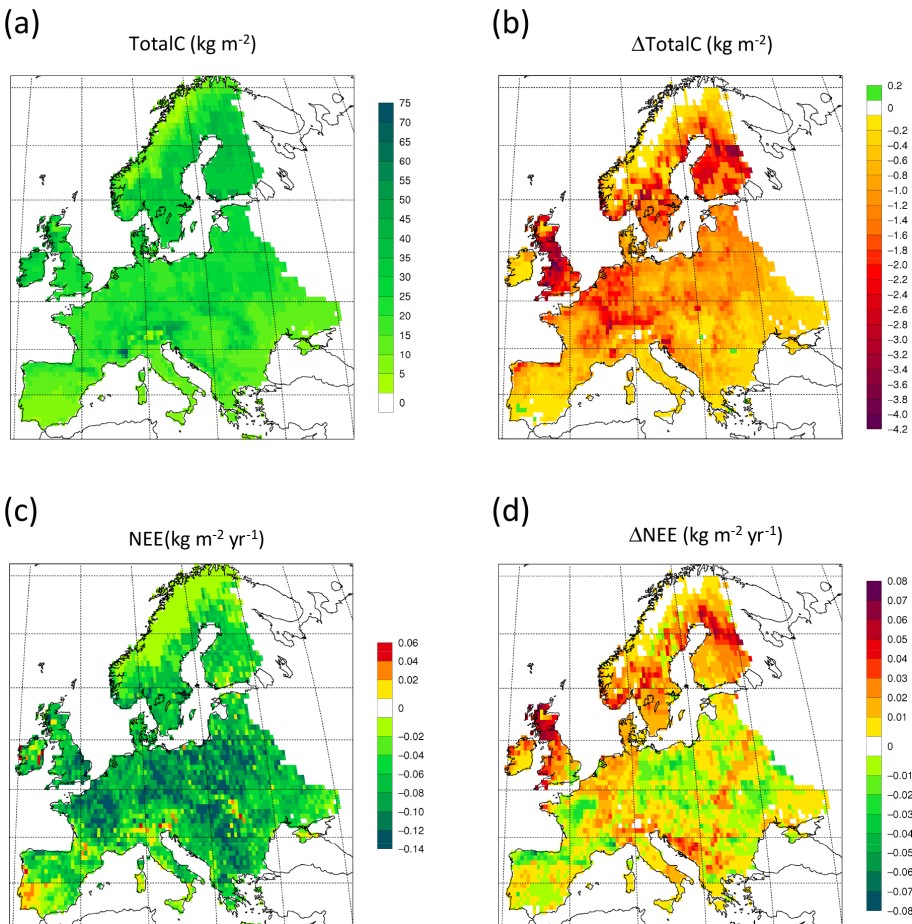

**Figure 14.** Simulated forest **(a)** total carbon pool 2010 in a simulation with thinning. **(b)** Total carbon pool 2010 difference between simulations with and without wood harvest in regrowth forest. **(c)** Mean 2001–2010 NEE in a simulation with thinning. **(d)** Mean 2001–2010 NEE difference between simulations with and without thinning.

ulations, as the optimum harvested volume required to completely avoid self-thinning may not be realised in real managed forest stands. Ideally, automated thinning should be just enough to avoid self-thinning mortality in the model, so the biomass should not be severely reduced, but in old beech stands self-thinning is very low in the model (Fig. C1), and thus in these stands both detailed and automated harvests result in a relatively large reduction in biomass compared to unharvested stands (Fig. 7).

The modelled mean vegetation carbon density in European forests in 2000–2010 is close to observations from several published sources (Pan et al., 2011; Liu et al., 2015; Forest Europe, 2015). Including thinning in the simulation has a rather small impact on vegetation carbon ($<5\%$), but after clear-cutting starts in regrowth stands after 2010, simulations with and without harvest in regrowth stands diverge strongly (Fig. E7). In addition, the modelled mean European growing stock is close to observations. Modelled carbon sink density ($= -$NEE) for European forests in a simulation without thinning in the present study is about 46 % of the 2000–2007 value reported by Pan et al. (2011). This is similar to the global carbon sink predicted by a simulation with a similar setup without thinning, which is 49 % of the global value from the Pan et al. (2011) study. The difference in modelled carbon sink in 2001–2010 between old-growth forest ($0.04\,\mathrm{kg\,C\,m^{-2}\,yr^{-1}}$) and regrowth forest without thinning ($0.085\,\mathrm{kg\,C\,m^{-2}\,yr^{-1}}$) is similar to the difference reported for global old-growth and regrowth forests by Pugh et al. (2019). Adding thinning to the European regrowth forest setup increases the carbon sink by 38 % for the total forest area and by 46 % for the regrowth forest area, reaching 64 % and 82 %, respectively, of the Pan et al. (2011) value. Thinning reduces natural mortality due to relaxed competition between trees, and since a large part of harvested biomass is removed from forest stands, litter input to the soil and the resulting heterotrophic respiration are also reduced (Figs. E6–E7), increasing the carbon sink.

Details in the simplified European setup might explain the remainder of the "missing" carbon sink relative to reported values. One potential cause is that old-growth ($>140$ year)

**Table 4.** Modelled and observed growing stock (GS) in European forests in 2010 and net annual increment (NAI) and fellings in forests available for wood supply (FAWS) in Europe for 2001–2010.

| | LPJ-GUESS[1] | Forest Europe[2] |
|---|---|---|
| **GS (million m$^3$)** | | |
| Europe | 38 136 (39 859) | 31 225 |
| EU-28 + Switzerland | 31 794 (33 385) | 25 357 |
| **GS (m$^3$ ha$^{-1}$)** | | |
| Europe | 156 (163) | 157 |
| EU-28 + Switzerland | 163 (171) | 158 |
| **NAI (million m$^3$ yr$^{-1}$)** | | |
| Europe | 966 (484) | 841 |
| EU-28 + Switzerland | 781 (401) | 732 |
| **NAI (m$^3$ ha$^{-1}$ yr$^{-1}$)** | | |
| Europe | 5.4 (2.7) | 5.1 TS3 |
| EU-28 + Switzerland | 5.4 (2.8) | 5.4 |
| **Fellings (million m$^3$ yr$^{-1}$)** | | |
| Europe | 896 (380) | 562 |
| EU-28 + Switzerland | 746 (333) | 527 |
| **Fellings (m$^3$ ha$^{-1}$ yr$^{-1}$)** | | |
| Europe | 5.0 (2.0) TS4 | 3.4 TS5 |
| EU-28 + Switzerland | 5.2 (2.3) | 3.9 |

[1] Mean of the years 2001–2010. Vegetation carbon to wood volume conversion factors for separate countries were derived from Forest Europe data. Values in brackets are for a simulation without wood harvest in regrowth forest. Regrowth forest in 2010 is considered FAWS in LPJ-GUESS simulations. [2] Mean of the years 2000, 2005, and 2010 or for the available data for these years, except for Greece NAI (1990 value). NAI value for Macedonia and fellings values for Belarus and Luxembourg are missing. The "Europe" area is the sum of the country areas.

forests in this study are represented by unmanaged PNV (with a low carbon sink; see Table 6), as in Pugh et al. (2019), missing effects of land-use history in Europe but preferred by us to the alternative of introducing arbitrary assumptions of age structure. Furthermore, the GFAD >140 year forest type data contain artefacts manifested in the BE distribution. Including a basic extensive wood harvest method in old-growth forests increased the total carbon sink by only 4 %, resulting in a value of 66 % of the Pan et al. (2011) value. Wildfires also contribute to a lower carbon sink in modelled PNV. A further likely cause of the discrepancy between the modelled and reported carbon balance is that secondary forests are created from PNV stands without taking land-use history into account. Reforestation of cropland, which generally has a much lower soil carbon content than forests in Europe (Guo and Gifford, 2002), has a higher carbon storage potential than regrowth after clearing of existing forests. In addition, soils of existing European forests have been depleted of carbon historically because of higher harvest rates, fuel-wood collection, and litter raking (Ciais et al., 2008; McGrath et al.,

2015). Higher initial soil carbon pools will increase the release of $CO_2$ in regrowth forests, especially under rising temperatures. Alternative methods to initialise secondary forests (fate of cleared wood, land-use history) have large implications for simulated carbon pools and fluxes as seen in the example Swedish site in this study, e.g. a mean carbon sink over 150 years spanning from 0.078 to 0.188 kg C m$^{-2}$ yr$^{-1}$ (Fig. 2). This has also been shown at the global scale (Pugh et al., 2019). The high value of modelled European soil carbon density in 2000–2010 (34 %–80 % higher than reported values) supports the possibility that the lack of consideration of LULCC history is a main source of the missing carbon sink in this study. The similarity of the modelled mean NAI of European forests in a simulation with thinning to observed values (a 100 % increase compared to a simulation without thinning) also suggests that the missing carbon sink component could be found in heterotrophic respiration and not in vegetation productivity.

The automated thinning and clear-cutting modelling strategy applied in the model in the present study is intended as an example for demonstrating the new forest management capabilities and an improvement on the age structure setup of Pugh et al. (2019) and does not include all available possibilities in the model. In addition to the shortcomings in the setup already noted concerning land-use history, many central European forests are managed by continuous wood harvest and not by clear-cutting and also consist of species mixes (Pretzsch et al., 2021). Estimating the effect of such different wood harvest strategies and monoculture or mixed-species alternatives on carbon stocks and fluxes is now possible and will be done in further studies. The self-thinning and tree-density-based harvest method is less successful in the northernmost and southernmost parts of Europe, where productivity is strongly limited by temperature and precipitation, respectively, and the self-thinning relationship between biomass and tree density in the model is weaker. The low simulated productivity of forests in the Mediterranean points to the need for a review of the parameterisation of tree species to reflect Mediterranean managed forests or the introduction of tree species that are not currently represented in the model (Fig. E8). While the model shows good skills when reproducing reported mean values for Europe's vegetation carbon and productivity, the correlation between modelled results and observations for the individual countries shows a large spread with no simple pattern for the deviations (Figs. E1–E5). However, it is obvious that modelled thinning intensities for countries in the Balkans, except Albania and Greece, are higher than the corresponding reported total harvest intensities. These countries also show a poorer fit to observed NAI values in a simulation with thinning compared to a simulation without thinning. In any case, including thinning in simulations improves the fit to observed national NAI values in most other countries.

Our simulation results using LPJ-GUESS exhibit similarity with results from the ORCHIDEE DVM, which was ap-

**Table 5.** Vegetation carbon and total carbon stock in European forests separated into regrowth and old-growth forest.

| | Total forest[1] | Regrowth forest[1] | Old-growth forest |
|---|---|---|---|
| **Veg C (Pg)** | | | |
| 2000 | 13.8 (14.3) | 6.6 (7.1) | 7.2 |
| 2007 | 14.1 (14.7) | 7.8 (8.3) | 6.4 |
| 2010 | 14.3 (15.0) | 8.3 (9.0) | 6.0 |
| 2015 | 14.2 (15.8) | 8.2 (9.8) | 6.1 |
| **Veg C (kg m$^{-2}$)** | | | |
| 2000 | 5.5 (5.7) | 4.0 (4.3) | 8.5 |
| 2007 | 5.7 (5.9) | 4.4 (4.7) | 8.8 |
| 2010 | 5.7 (6.0) | 4.5 (4.9) | 9.1 |
| 2015 | 5.7 (6.4) | 4.5 (5.3) | 9.2 |
| **Soil + Litter C (Pg)** | | | |
| 2000 | 46.5 (47.6) | 30.9 (32.4) | 15.6 |
| 2007 | 46.3 (48.1) | 33.1 (34.9) | 13.2 |
| 2010 | 46.2 (48.2) | 34.0 (36.0) | 12.2 |
| 2015 | 46.1 (48.1) | 34.0 (35.9) | 12.2 |
| **Soil + Litter (kg m$^{-2}$)** | | | |
| 2000 | 18.6 (19.2) | 18.8 (19.6) | 18.4 |
| 2007 | 18.5 (19.3) | 18.6 (19.6) | 18.3 |
| 2010 | 18.5 (19.3) | 18.6 (19.7) | 18.3 |
| 2015 | 18.5 (19.3) | 18.6 (19.6) | 18.3 |
| **Total C stock (Pg)** | | | |
| 2000 | 60.3 (62.3) | 37.5 (39.5) | 22.7 |
| 2007 | 60.4 (62.8) | 40.9 (43.2) | 19.5 |
| 2010 | 60.5 (63.1) | 42.3 (45.0) | 18.2 |
| 2015 | 60.6 (64.0) | 42.1 (45.7) | 18.2 |
| **Total C stock (kg m$^{-2}$)** | | | |
| 2000 | 24.2 (25.0) | 22.8 (24.0) | 26.9 |
| 2007 | 24.2 (25.2) | 23.0 (24.3) | 27.2 |
| 2010 | 24.3 (25.3) | 23.1 (24.6) | 27.4 |
| 2015 | 24.3 (25.6) | 23.0 (25.0) | 27.5 |

[1] Values in parentheses are for a simulation without wood harvest in regrowth forest. Harvest products were not included in the calculations of total carbon. Total Europe area definition is as in Table 2.

plied with the same automated thinning method at a central European site (Bellassen et al., 2010). The ORCHIDEE simulation with automated thinning, compared to a simulation without thinning, gave a similar vegetation reduction (7 %) and thinning fraction (0.55), reduced heterotrophic respiration (ca. 20 %), and a carbon sink increase (67 %). The forest NPP reduction over time in ORCHIDEE simulations (ca. 10 %) is also seen in the average value for unharvested regrowth forests in European simulations with LPJ-GUESS (Fig. E7b). The decline of NPP directly after thinnings in ORCHIDEE is not simulated by LPJ-GUESS, but both models display a short-lived increase in heterotrophic respiration after thinnings (not shown). The recovery time after a clear-cut (when the stand turns into a carbon sink) is 6 years in

the example southern Swedish site with a standard harvest removal, but it is 18 years if the harvested biomass is left on site (Fig. 2). This is similar to the ORCHIDEE results with a stand recovery time of 10–20 years after a clear-cut. A similar recovery time after clear-cutting, 7–11 years, has been diagnosed based on $CO_2$ flux measurements in Sweden (Lindroth et al., 2009).

Responses of soil carbon and nitrogen cycling to harvest and fertilisation can be complex and qualitatively different in clear-cut and continuous-harvest systems (Parolari and Porporato, 2016). The coupled carbon–nitrogen cycling in LPJ-GUESS (Smith et al., 2014) should enable the investigation of the effect of different management practices on forest productivity and sustainability at both stand and regional scale in

**Table 6.** Net ecosystem exchange (NEE), harvested carbon, and natural mortality in European forests[1] separated into regrowth and old-growth forest.

| | Total forest | Regrowth forest | Old-growth forest |
|---|---|---|---|
| NEE (Pg C yr$^{-1}$) | | | |
| 1991–2000 | −0.187 (−0.140) | −0.158 (−0.111) | −0.028 |
| 2000–2007 | −0.212 (−0.153) | −0.188 (−0.129) | −0.024 |
| 2001–2010 | −0.234 (−0.178) | −0.204 (−0.148) | −0.030 |
| 2011–2015 | −0.211 (−0.159) | −0.200 (−0.148) | −0.011 |
| NEE (kg C m$^{-2}$ yr$^{-1}$) | | | |
| 1991–2000 | −0.075 (−0.056) | −0.106 (−0.072) | −0.030 |
| 2000–2007 | −0.085 (−0.061) | −0.110 (−0.075) | −0.031 |
| 2001–2010 | −0.094 (−0.071) | −0.117 (−0.085) | −0.040 |
| 2011–2015 | −0.085 (−0.064) | −0.109 (−0.081) | −0.016 |
| Harvest (Pg C yr$^{-1}$) | | | |
| 1991–2000 | 0.196 (0.102) | 0.094 (0) | 0.102 |
| 2001–2010 | 0.210 (0.093) | 0.117 (0) | 0.093 |
| 2011–2015 | 0.241 (0) | 0.241 (0) | 0 |
| Harvest (kg C m$^{-2}$ yr$^{-1}$) | | | |
| 1991–2000 | 0.079 (0.041) | 0.061 (0) | 0.109 |
| 2001–2010 | 0.084 (0.037) | 0.067 (0) | 0.125 |
| 2011–2015 | 0.097 (0) | 0.132 (0) | 0 |
| Mortality (Pg C yr$^{-1}$) | | | |
| 1991–2000 | 0.104 (0.201) | 0.025 (0.123) | 0.079 |
| 2001–2010 | 0.099 (0.227) | 0.032 (0.159) | 0.067 |
| 2011–2015 | 0.100 (0.240) | 0.035 (0.176) | 0.064 |
| Mortality (kg C m$^{-2}$ yr$^{-1}$) | | | |
| 1991–2000 | 0.042 (0.081) | 0.016 (0.079) | 0.084 |
| 2001–2010 | 0.040 (0.091) | 0.018 (0.091) | 0.090 |
| 2011–2015 | 0.040 (0.096) | 0.019 (0.096) | 0.096 |

[1] Values in parentheses are for a simulation without wood harvest in regrowth forest. Total Europe area definition is as in Table 2.

future studies. Nitrogen depletion of the soil in previous land-use history reduces forest productivity and causes a shift in species succession in the model (Fig. 2c). At the European scale, removing a smaller fraction of residues (0 % of leaves rather than 30 %) makes a small positive impact on productivity (0.1 %; see Sect. 3.4). However, since many European forests receive large amounts of atmospheric nitrogen deposition, other nutrients such as Ca, Mg, K, and P may be more important for limiting productivity, and acidification of the soil by N and S deposition may further decrease the availability of these nutrients (Sverdrup et al., 2006). Ca is especially close to or below the limit of sustainability in current forest management systems in southern Sweden (Sverdrup et al., 2006). Thus, ongoing development of limitation and cycling of additional nutrient species into LPJ-GUESS may be beneficial for capturing the full effects of different harvest

regimes. Also relevant to achieving a better model of nutrient uptake is an improved representation of the soil profile.

While the mean productivity of European forests is captured well by the model (Table 4), and mean productivity of forests in individual European countries is captured reasonably well (Figs. 10, E4), the inability to reproduce observed productivity levels in high-productivity beech and spruce stands in Germany (Fig. 7a) highlights the need for allowing a wider range of productivities. The lack of certain physiological processes in the model, e.g. hardening and dehardening (Bergh et al., 1998), could explain why productivities along the whole temperature gradient in European forests cannot be fully represented in the model. Model tuning that aims for correct mean values of, for example, biomass and carbon fluxes over large geographic areas compensates for an overestimation of productivity in northern Europe by low-

ering average productivity along the whole temperature gradient. This could partially explain, for example, why the productivity of some southern German sites is underestimated, while average productivity for Germany as a whole is in line with inventory data. Additionally, the selected individual German Norway spruce and European beech sites in this study were generally of above-average site quality and are not fully representative of German forests, which includes forests of other tree species, especially Scots pine (*Pinus sylvestris*) and oak species (*Quercus robur*/*Quercus petraea*), on lower site quality sites. This is likewise in line with the smaller gap between modelled and observed growing stock (ca. 20 %, Fig. E3) seen at country level, compared to individual spruce and beech sites in Germany (Fig. 7a).

The emergent competition between PFTs with similar shade-tolerance values in the model, e.g. beech and spruce, can deviate from actual dynamics, as seen in the poor performance of spruce compared to beech in a succession at the example site in southern Sweden (Fig. 5).

The management systems covered by the new forest management functionality in LPJ-GUESS include the most important features required for the improvement of modelling carbon pools and fluxes and the development of forest stands under future climates, but a few important additions will be desirable to include in the future. These include automated continuous wood harvesting and coppice management. For a good representation of coppicing, the model should also be improved to include plant carbohydrate storage. For better representations of European forests, land-use history, including litter raking, should be included to generate more realistic soil carbon pools by adapting functionality already available in the model.

## Appendix A:  Supplementary model parameterisation tables

**Table A1.** PFT parameters used in this study. Values in bold text are updated compared to Hickler et al. (2012).

| Species/PFT | Phenology | Geographic range[1] | Shade tolerance[1] | Growth form[1] | $Tc_{min}$ | $Tc_{max}$ | $Tw_{min}$ | $GDD_5$ |
|---|---|---|---|---|---|---|---|---|
| *Abies alba* | EG | temperate | tolerant | tree | **−6.5 (−7.5)** | **2** | 6 | **1600** |
| *Betula pendula* | SG | temperate | intolerant | tree | −30 | **7** | 5 | 700 |
| *Betula pubescens* | SG | boreal | intolerant | tree | −30 | **3** | 5 | 350 |
| *Carpinus betulus* | SG | temperate | intermediate | tree | −8 | **5** | 5 | 1200 |
| *Corylus avellana* | SG | temperate | intermediate | tree | **−11** | **7** | 5 | 800 |
| *Fagus sylvatica* | SG | temperate | tolerant | tree | **−6 (−8)** | **6** | 5 | 1500 |
| *Fraxinus excelsior* | SG | temperate | intermediate | tree | −16 | **6** | 5 | 1100 |
| *Juniperus oxycedrus* | EG | temperate | intolerant | **tree** | 1 (0) | – | – | 2200 |
| ***Larix decidua*** | SG | boreal | intermediate | tree | −30 | −2 | 5 | 300 |
| *Picea abies* | EG | boreal | tolerant | tree | −30 | −1.5 | 5 | 600 |
| *Pinus halepensis* | EG | temperate | intolerant | tree | 3 | **9** | 21 | 3000 |
| *Pinus sylvestris* | EG | boreal | intermediate | tree | −30 | −1 | 5 | 500 |
| ***Populus tremula*** | SG | temperate | intolerant | tree | **−30 (−31)** | **6** | – | 500 |
| *Quercus coccifera* | EG | temperate | intermediate | shrub | 0 | **11** | 21 | 2200 |
| *Quercus ilex* | EG | temperate | intolerant | tree | **3** | **7** | 5 | 1800 |
| *Quercus pubescens* | SG | temperate | intermediate | tree | −5 | **6** | – | 1900 |
| *Quercus robur* | SG | temperate | intermediate | tree | **−9 (−10)** | **6** | 5 | 1100 |
| *Tilia cordata* | SG | temperate | intermediate | tree | **−11 (−12)** | **5** | 5 | **1100** |
| ***Ulmus glabra*** | SG | temperate | intermediate | tree | −9.5 (−10.5) | **6** | 5 | 850 |
| Boreal evergreen shrub | EG | boreal | intolerant* | shrub | – | −1 | – | **200** |
| Mediterranean raingreen shrub | RG | temperate | intolerant | shrub | 1(0) | – | – | 2200 |
| $C_3$ grass | SG/RG | temp-boreal | – | herb | – | – | – | – |

| Species/PFT | $k_{allom1}$ | $k_{la:sa}$ | gmin ($mm\,s^{-1}$) | Chilling requirement[1] | fAWC | $CA_{max}$ ($m^2$) | $z_1$ | $r_{fire}$ | $\alpha_{leaf}$ (yr) | $\alpha_{ind}$ (yr) | fnstorage |
|---|---|---|---|---|---|---|---|---|---|---|---|
| *Abies alba* | 150 | 4000 | 0.3 | – | 0.35 | 40 | **0.6** | 0.1 | 3 | 350 | 0.05 |
| *Betula pendula* | 250 | 5000 | 0.5 | intermediate | 0.42 | 40 | **0.6** | 0.1 | 0.5 | 200 | 0.15 |
| *Betula pubescens* | 250 | 5000 | 0.5 | intermediate | 0.5 | 40 | **0.6** | 0.1 | 0.5 | 200 | 0.15 |
| *Carpinus betulus* | 250 | 5000 | 0.5 | high | 0.33 | 40 | **0.6** | 0.1 | 0.5 | 350 | 0.15 |
| *Corylus avellana* | 250 | 4000 | 0.5 | intermediate | 0.3 | 40 | **0.6** | 0.1 | 0.5 | **100** | 0.15 |
| *Fagus sylvatica* | 250 | 5000 | 0.5 | high | 0.3 | 40 | **0.6** | 0.1 | 0.5 | 500 | 0.15 |
| *Fraxinus excelsior* | 250 | 5000 | 0.5 | low | 0.4 | 40 | **0.6** | 0.1 | 0.5 | 350 | 0.15 |
| *Juniperus oxycedrus* | 150 | 1500 | 0.5 | – | 0.01 | 10 | 0.5 | 0.4 | 1.5 | 200 | 0.05 |
| ***Larix decidua*** | 150 | 5000 | 0.3 | low | 0.3 | 40 | 0.6 | 0.2 | 1 | 500 | 0.05 |
| *Picea abies* | 150 | 4000 | 0.3 | – | 0.43 | 40 | 0.8 | 0.1 | **3** | 500 | 0.05 |
| *Pinus halepensis* | 150 | **3000** | 0.3 | – | 0.05 | 40 | 0.6 | 0.2 | 2 | 350 | 0.05 |
| *Pinus sylvestris* | 150 | **3000** | 0.3 | – | 0.25 | 40 | 0.6 | 0.2 | 2 | **350** | 0.05 |
| ***Populus tremula*** | 250 | 5000 | 0.5 | intermediate | 0.4 | 40 | 0.7 | 0.2 | 0.5 | 160 | 0.15 |
| *Quercus coccifera* | **100** | 2500 | 0.5 | – | 0.1 | 10 | 0.5 | 0.3 | 1.5 | 350 | 0.3 |
| *Quercus ilex* | 250 | 3000 | 0.5 | – | 0.1 | 40 | 0.5 | 0.3 | 2 | 350 | 0.05 |
| *Quercus pubescens* | 250 | 5000 | 0.5 | low | 0.2 | 40 | 0.6 | 0.2 | 0.5 | 500 | 0.15 |
| *Quercus robur* | 250 | 5000 | 0.5 | low | 0.25 | 40 | 0.6 | 0.2 | 0.5 | 500 | 0.15 |
| *Tilia cordata* | 250 | 5000 | 0.5 | high | 0.33 | 40 | 0.8 | 0.1 | 0.5 | 350 | 0.15 |
| ***Ulmus glabra*** | 250 | 5000 | 0.5 | low | 0.4 | 40 | 0.6 | 0.1 | 0.5 | 350 | 0.15 |
| Boreal evergreen shrub | **20** | 500 | 0.3 | – | 0.25 | 3 | 0.8 | 0.1 | 2 | 50 | 0.3 |
| Mediterranean raingreen shrub | **100** | 1500 | 0.5 | – | 0.01 | 10 | 0.9 | 0.3 | 0.5 | 100 | 0.3 |
| $C_3$ grass | – | – | 0.03 | – | 0.01 | – | 0.9 | 0.5 | 0.5 | – | 0.3 |

[1] See group parameter Table A2. Phenology is abbreviated as follows: evergreen (EG), summergreen (SG), and raingreen (RG). $Tc_{min}$, $Tc_{max}$ = minimum and maximum temperature of the coldest month for establishment, values in brackets are the minimum temperature for survival if this is different from the value for establishment. $Tw_{min}$ = minimum warmest month mean temperature for establishment. $GDD_5$ = minimum degree-day sum above 5 °C for establishment. $k_{allom1}$ = constant in allometry equations (Smith et al., 2001). $k_{la:sa}$ = leaf area to sapwood cross-sectional area ratio. gmin = minimum canopy conductance. fAWC = minimum growing-season (daily temperature >5 °C) fraction of available soil water holding capacity in the first soil layer. $CA_{max}$ = maximum woody crown area. $z_1$ = fraction of roots in first soil layer. $r_{fire}$ = fraction of individuals surviving fire. $a_{leaf}$ = leaf longevity. $a_{ind}$ = maximum, non-stressed longevity. fnstorage is the fraction of sapwood (root for herbaceous PFTs) that can be used as a nitrogen long-term storage scalar.

**Table A2.** Common PFT parameters for shade tolerance, geographic range, growth form, and chilling requirement categories in Table A1. Values in bold text are updated compared to Hickler et al. (2012).

| Shade tolerance | tolerant | intermediate | intolerant |
|---|---|---|---|
| Sapwood to heartwood conversion rate $(\text{yr}^{-1})$* | 0.05 | **0.075** | **0.1** |
| Growth efficiency parameter $(\text{kg C m}^{-2}\text{yr}^{-1})$ | 0.04 | **0.06** | **0.08** |
| Max. establishment rate (saplings $\text{yr}^{-1}\,\text{m}^{-2}$) | 0.05 | **0.15** | 0.2 |
| Min. PAR at forest floor for establishment $(\text{MJ m}^{-2}\,\text{d}^{-1})$ | **0.35** | 2.0 | 2.5 |
| Recruitment shape parameter | **3** | **7** | 10 |

| Geographic range | boreal | temperate | temperate–boreal grass |
|---|---|---|---|
| Base respiration rate at $10\,°C$ $(\text{g C g N}^{-1}\,\text{d}^{-1})$ | **1** | 1 | 1 |
| Optimum temperature range for photosynthesis (°C) | 10–25 | 15–25 | 10–30 |
| pstemp_min (°C) | −4 | −2 | −5 |
| pstemp_max (°C) | 38 | 38 | 45 |

| Growth form | tree | shrub | herbaceous |
|---|---|---|---|
| $k_{\text{allom2}}$ (allometric parameter) | 40 | 5 | – |
| Wood density $(\text{kg C m}^{-3})$ | 200 | 250 | – |
| $\text{lr}_{\text{max}}$ non-water-stressed leaf to fine-root mass ratio | 1 | 1 | 0.5 |
| Fine-root turnover rate $(\text{yr}^{-1})$ | 0.7 | 0.7 | 0.7 |

| Chilling requirement | low | intermediate | high |
|---|---|---|---|
| k_chilla | 0 | 0 | 0 |
| k_chillb | 100 | 350 | 600 |
| k_chillk | 0.05 | 0.05 | 0.05 |

* Boreal evergreen shrub: 0.05.

**Table A3.** Parameters for automated thinning and clear-cutting.

| | $\alpha_{\text{st}}$ (trees $\text{ha}^{-1}$) | $\beta_{\text{st}}$ log(trees $\text{ha}^{-1}$) (log $\text{m}^{-1}$) | $\text{rdi}_{\text{target}}$ | $\text{dens}_{\text{target}}$ (trees $\text{ha}^{-1}$) |
|---|---|---|---|---|
| Needleleaf (NL) | 65 | 1.6 | 0.7 | 250 |
| Broadleaf (BL) | 40 | 1.6 | 0.85 | 100 |

## Appendix B: Supplementary model initialisation and management options figures

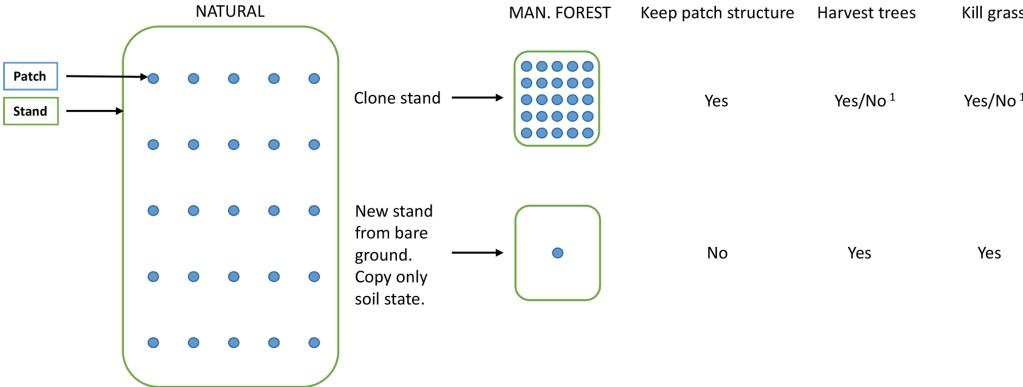

**Figure B1.** Options when creating managed forest stands from PNV. [1] For the cloning alternative, tree harvesting and grass killing are optional.

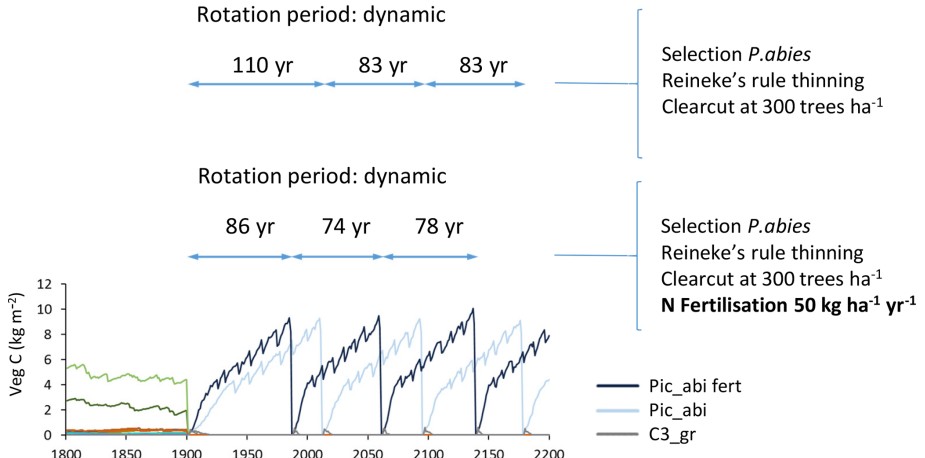

**Figure B2.** Effect of nitrogen fertilisation (50 kg ha$^{-1}$ yr$^{-1}$) on modelled productivity and rotation length in spruce monoculture with automated thinning and clear-cutting. Abbreviations are as follows: Pic_abi fert, *Picea abies* with N fertilisation; Pic_abi, *Picea abies* without N fertilisation; C3_gr, C$_3$ grass. Forestry stands were created from clear-cutting of PNV in 1901. Location, climate input, and species in PNV are as in Fig. 2.

## Appendix C: Supplementary European simulation setup tables and figures

**Table C1.** Mapping of EFI tree groups to LPJ-GUESS species selections.

| EFI species group[1] | LPJ-GUESS selection |
| --- | --- |
| Broadleaf deciduous (BD) | |
| Alnus, Betula | *B. pendula, B. pubescens* |
| BroadleafMisc, Castanea, Robinia | *B. pendula, B. pubescens, C. avellana, Q. pubescens, T. cordata, U. glabra* |
| Carpinus | *C. betulus* |
| Fagus | *F. sylvatica* |
| Fraxinus | *F. excelsior* |
| Populus | *P. tremula* |
| QuercusRobPet | *Q. robur* |
| None[2] | *B. pubescens, F. sylvatica, Q. robur, C. avellana* |
| Undet.[3] | *B. pendula, B. pubescens, C. betulus, C. avellana, F. sylvatica, F.excelsior, P. tremula, Q. pubescens, Q. robur, T. cordata, U.glabra* |
| Broadleaf evergreen (BE) | |
| QuercusMisc, Eucalyptus | *Q. ilex* |
| Needleleaf deciduous (ND) | |
| Larix | *L. decidua* |
| Needleleaf evergreen (NE) | |
| Abies | *A. alba* |
| Conifers, Pseudotsuga | *P. abies, P. sylvestris, P. halepensis* |
| Picea | *P. abies* |
| PinusSylv | *P. sylvestris* |
| PinusMisc, PinusPin | *P. sylvestris, P. halepensis* |
| None[2] | *P. abies, P. sylvestris* |
| Undet.[3] | *A. alba, P. abies, P. sylvestris* |

[1] Abbreviations of EFI species/species groups are as follows: Abies (*Abies* spp.), Alnus (*Alnus* spp.), BroadleafMisc (other broadleaves), Betula (*Betula* spp.), Carpinus (*Carpinus* spp.), Castanea (*Castanea* spp.), Conifers (other conifers), Eucalyptus (*Eucalyptus* spp.), Fagus (*Fagus* spp.), Fraxinus (*Fraxinus* spp.), Larix (*Larix* spp.), Picea (*Picea* spp.), PinusPin (*P. pinaster*), PinusSylv (*P. sylvatica*), PinusMisc (*Pinus* spp. other than *P. pinaster* and *P. sylvestris*)), Populus (*Populus* spp.), Pseudotsuga (*P. menziesii*), QuercusRobPet (*Q. robur, Q. petraea*), and Robinia (*Robinia* spp.). [2] Grid cells without EFI forest. [3] Undetermined equal fractions of all EFI tree groups.

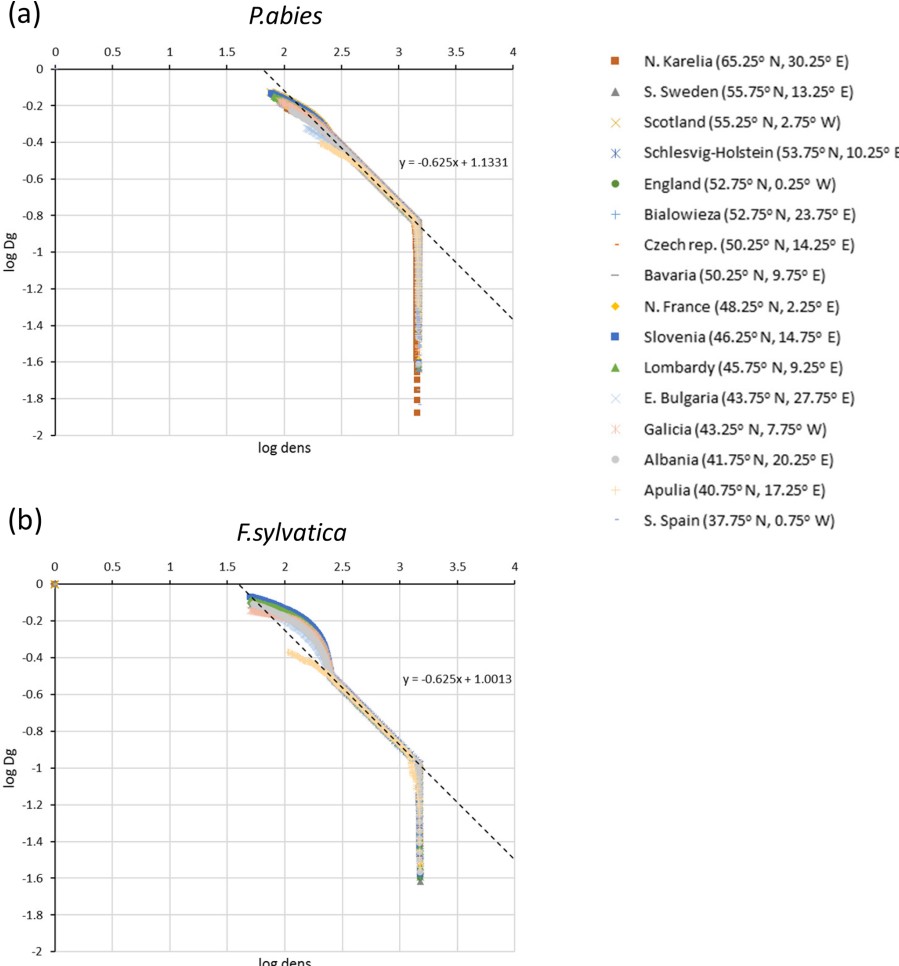

**Figure C1.** Self-thinning log–log plots of quadratic mean diameter (Dg) and tree density (dens) for simulations of **(a)** *Picea abies* and **(b)** *Fagus sylvatica* monocultures at 16 European sites used for automated thinning in the model.

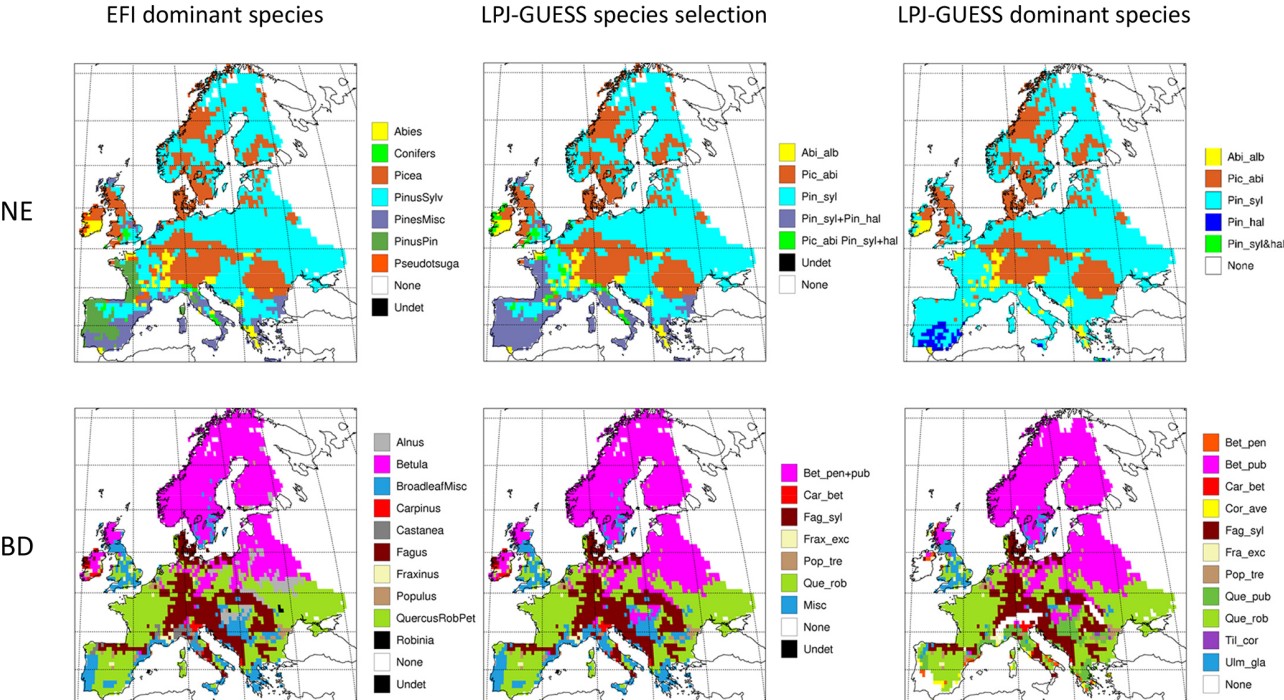

**Figure C2.** Mapping of dominant EFI tree species groups in the needleleaf evergreen (NE) and broadleaf deciduous (BD) GFAD forest classes to LPJ-GUESS species selections and the resulting dominant species (LAI) in 1986–2015 in an LPJ-GUESS simulation with automated thinning. Abbreviations of EFI species and species groups are as follows: Abies (*Abies* spp.), Alnus (*Alnus* spp.), BroadleafMisc (other broadleaves), Betula (*Betula* spp.), Carpinus (*Carpinus* spp.), Castanea (*Castanea* spp.), Conifers (other conifers), Eucalyptus (*Eucalyptus* spp.), Fagus (*Fagus* spp.). Fraxinus (*Fraxinus* spp.), Larix (*Larix* spp.), Picea (*Picea* spp.), PinusPin (*P. pinaster*), PinusSylv (*P. sylvatica*), PinusMisc (*Pinus* spp. other than *P. pinaster* and *P.sylvestris*), Populus (*Populus* spp.), Pseudotsuga (*P. menziesii*), QuercusRobPet (*Q. robur*, *Q. petraea*), Robinia (*Robinia* spp.). Abbreviations of LPJ-GUESS species and species groups are as follows: Abi_alb (*A.alba*), Pic_abi (*P.abies*), Pin_ syl (*P.sylvestris*), Pin_hal (*P.halipensis*), Pin_ syl+hal (*P.sylvestris+P.halipensis*), Bet_pen (*B.pendula*), Bet_pub (*B.pubescens*), Bet_pen+pub (*B.pendula+B.pubescens)*, Car_bet (*C.betulus*), Cor_ave (*C.avellana*), Fag_syl (*F.sylvestris*), Frax_exc (*F.excelsior*), Pop_tre (*P.tremul*a), Que_rob (*Q.robur*), Que_pub (*Q.pubescens*), Til_cor (*T.cordata*), Ulm_gla (*U.glabra*). The EFI groups BroadleafMisc, Castanea, and Robinia are mapped to the LPJ-GUESS selection "Misc", including the following species: *B.pendula*, *B.pubescens*, *C.avellana*, *Q.pubescens*, *T.cordata*, and *U.glabra*. For the mapping of the EFI groups None and Undet, see Table C1.

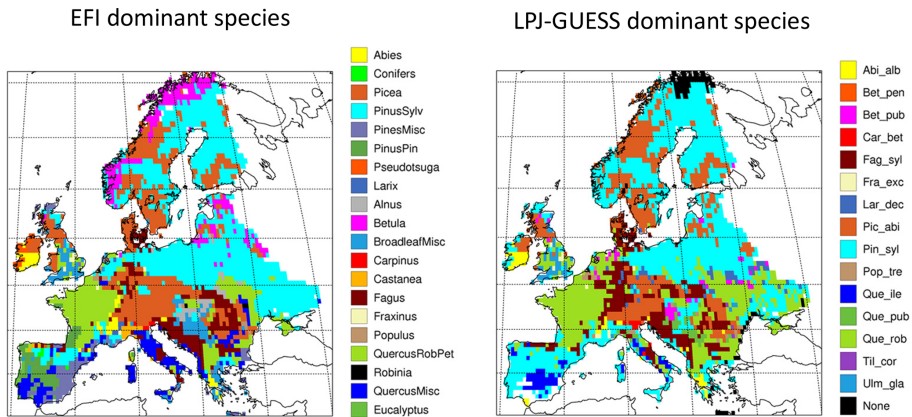

**Figure C3.** Comparison of dominant EFI tree species groups (area) and modelled LPJ-GUESS managed forest dominant tree species (LAI) in 1986–2015 in an LPJ-GUESS simulation with automated thinning. Abbreviations of LPJ-GUESS species are as in Fig. C2.

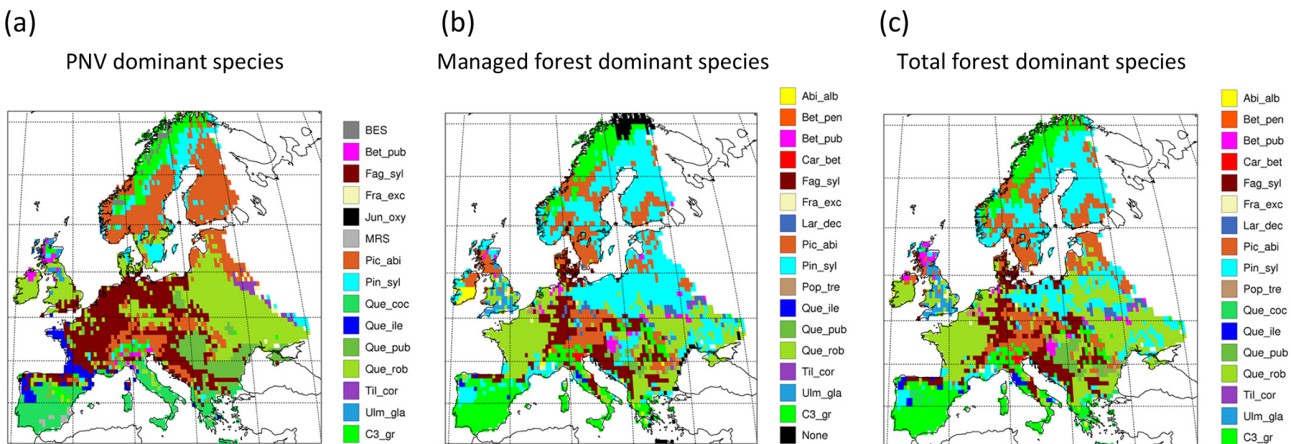

**Figure C4.** Modelled LPJ-GUESS dominant species (LAI) (including grass) in **(a)** primary forest (modelled as PNV), **(b)** secondary forest (managed with automated thinning), and **(c)** the total forest landscape in 1986–2015. Abbreviations of LPJ-GUESS tree species are as in Fig. C2: BES (boreal evergreen shrub), MRS (Mediterranean raingreen shrub), and C3_gr (C3 grass).

## Appendix D: Supplementary central European site information

The GER-Bav dataset contains pure European beech (three sites) and pure Norway spruce (three sites) and comes from the database of the Chair of Forest Growth and Yield Science TUM School of Life Sciences Technical University of Munich. Mean annual temperature is 6–7.7 °C, mean annual precipitation is 800–1200 mm, and elevation is 400–820 m a.s.l. Site quality is average to very good. Applied management is light, moderate, or heavy thinning.

The GER-C dataset contains pure European beech stands (three sites) and pure Norway spruce stands (five sites) and comes from the database of long-term research plots from Nordwestdeutsche Forstliche Versuchsanstalt, Abteilung Waldwachstum. Site quality is from average to above average, mean annual temperature is 6.5–8.5 °C, mean annual precipitation is 730–1100 mm, and elevation is 310–610 metres above sea level (m a.s.l.). Thinning methods were thinning from above, thinning from below (light, moderate, and heavy), and selective thinning.

The GER-CS dataset (Pretzsch, 2005; Pretzsch and Biber, 2005) is derived from long-term thinning experiments in pure stands of Norway spruce (eight sites) and European beech (nine sites), mostly in the lowlands or subalpine parts of southern and central Germany. Plot sizes were 0.25–0.5 ha. The spruce plots were concentrated on the southern German Pleistocene in the natural habitat of Norway spruce and were artificially established in re-afforestation after clear-cutting or afforestation of cropland and pastures. The site fertility was excellent (class I and II). The plots were subjected to light, moderate, and heavy thinning as was also the case for the GER-Bav dataset. The beech plots represented sites with average to very good fertility on red marl and red sandstone soils in central Germany and were the result of natural regeneration following cutting according to a compartment shelterwood system, resulting in consistently even-aged stands despite natural regeneration. For the beech plots, mean annual temperature is 6.5–8.8 °C, mean annual precipitation is 660–1080 mm, and elevation is 310–610 m a.s.l. For the spruce plots, mean annual temperature is 6.2–8 °C, the mean sum of annual precipitation is 1010–1200 mm, and the elevation is 340–840 m a.s.l. The main thinning method was thinning from below with thinning intensity grades $A$, $B$, and $C$ that correspond to light, moderate, and heavy thinning, respectively, and are defined according to the Association of German Forest Research Stations (Verein Deutscher Forstlicher Versuchsanstalten, 1902) and described by Pretzsch (2005).

The SLO dataset consisted of 27 forest sub-compartments of an average size of 25.6 ha from the high karst plateau Pokljuka in the Alps (46.35° N, 13.96° E, Slovenia, 1312 m a.s.l.). The area is characterised by pure Norway spruce even-aged stands in the timber phase (on average $120 \pm 20$ years old and with a growing stock of $568 \pm 118$ m$^3$ ha$^{-1}$). The climate is alpine with n annual range of precipitation of 1900 to 2300 mm and mean annual temperature of 3 °C. Site productivity is around 8 m$^3$ ha$^{-1}$. The forests are now parts of the Triglav National Park but were intensively harvested in the 18th and 19th centuries for the iron industry using clear-cutting and shelterwood systems. The current forest management system is a combination of various shelterwood and group-selection systems. In the last 30 years, mean decadal harvesting intensities in the selected sub-compartments were 14 % of the growing stock.

**Table D1.** Central European beech and Norway spruce site data used in the study.

| Dataset | Location | Source | No. stands | No. sites | Stand age (yr) | Last sampling year | No. of samplings | Mean sampling interval (yr) | Replicate stands[1] | Mean harvest intensity[1] |
|---|---|---|---|---|---|---|---|---|---|---|
| Beech | | | | | | | | | | |
| GER-Bav | Bavaria | This paper | 4 | 3 | 44–139 | 2012–2014 | 5–10 | 5–7 | 1–2 | 0.05–0.154 |
| GER-C | Central Germany | Ralf Nagel, personal communication, 2019 | 6 | 3 | 35–169 | 2014–2015 | 6–21 | 5–6 | 2 | 0.014–0.033, 0.098–0.134 |
| GER-CS | Central and southern Germany | Pretzsch (2005) | 27 | 9 | 100 | 1905–1995 | 1 | 100 | 3 | 0.086–0.213, 0.294–0.392, 0.396–0.595 |
| Spruce | | | | | | | | | | |
| GER-Bav | Bavaria | This paper | 3 | 3 | 30–105 | 2013–2016 | 7–11 | 5–7 | 1 | 0.075–0.152 |
| GER-C | Central Germany | Ralf Nagel, personal communication, 2019 | 9 | 5 | 23–124 | 2005–2018 | 4–19 | 5 | 1–2 | 0.006–0.038, 0.063–0.149 |
| GER-CS | Central and southern Germany | Pretzsch (2005) | 26 | 9 | 100 | 1947–1986 | 1 | 100 | 2–3 | 0.265–0.357, 0.303–0.433, 0.316–0.518 |
| SLO | Slovenia | This paper | 27 | 1 | 24–145 | 2015 | 4 (3)[2] | 10 | 1 | 0.039–0.249 |

[1] Mean harvest intensity range of one, two, or three different thinning intensities in replicate stands during a sampling interval. [2] Harvest reported for the last three observations only.

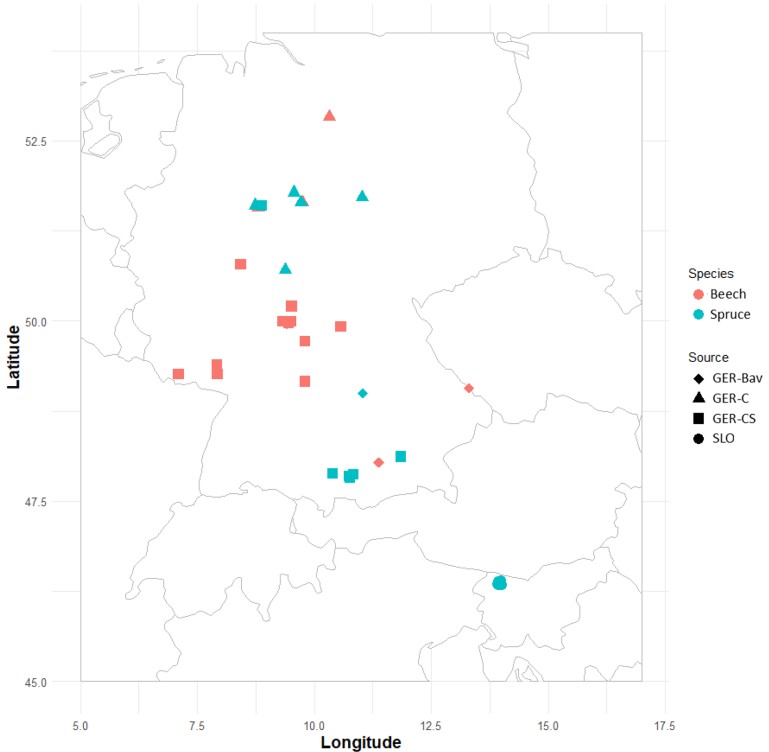

**Figure D1.** Location of the beech and spruce sites for the four stand datasets.

## Appendix E: Supplementary European simulation evaluation figures

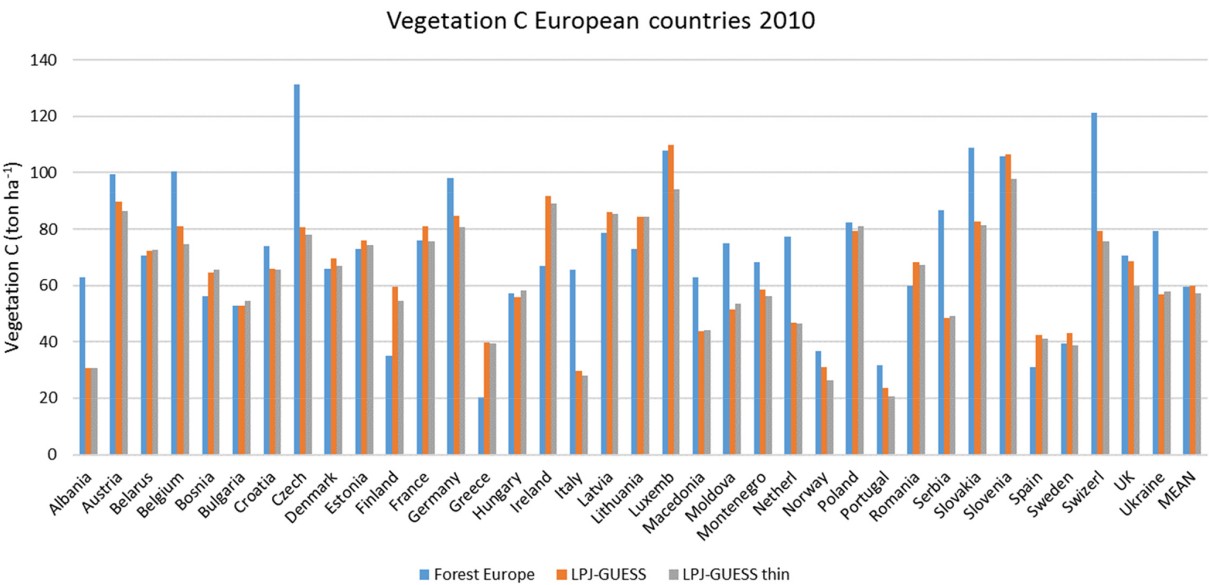

**Figure E1.** Modelled and observed (Forest Europe, 2015) vegetation carbon for individual countries in 2010. LPJ-GUESS is the simulation without thinning. LPJ-GUESS thin is the simulation with automated thinning.

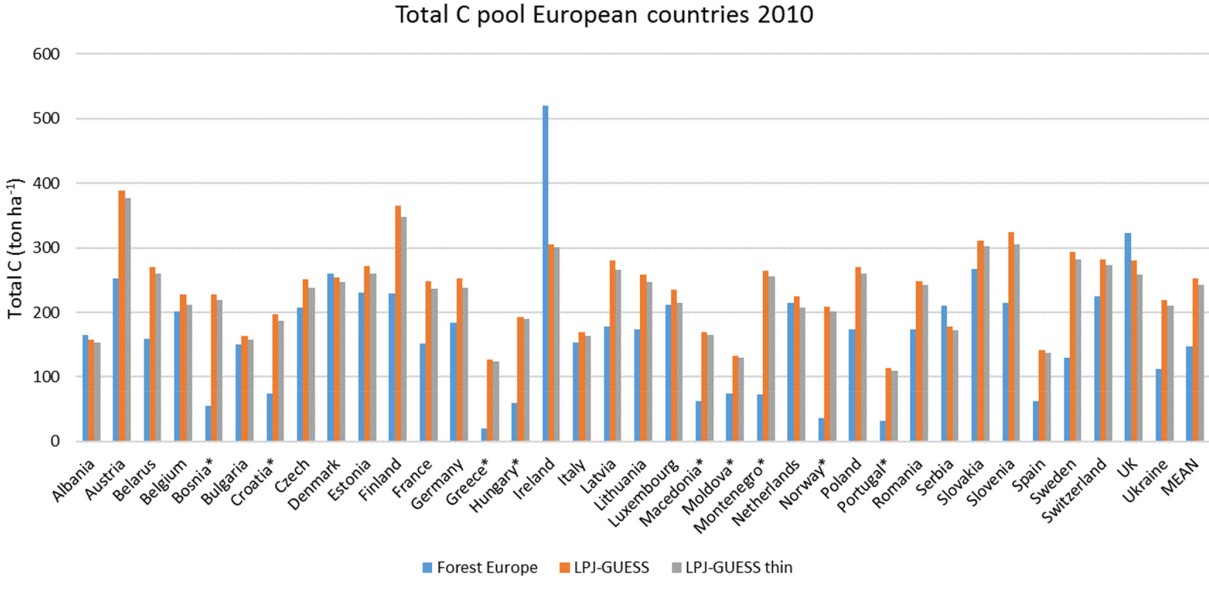

**Figure E2.** Modelled and observed (Forest Europe, 2015) total carbon pool for individual countries in 2010. LPJ-GUESS is the simulation without thinning. LPJ-GUESS thin is the simulation with automated thinning. * Soil and litter carbon data are missing for Bosnia, Croatia, Greece, Hungary, Macedonia, Moldova, Montenegro, Norway, and Portugal.

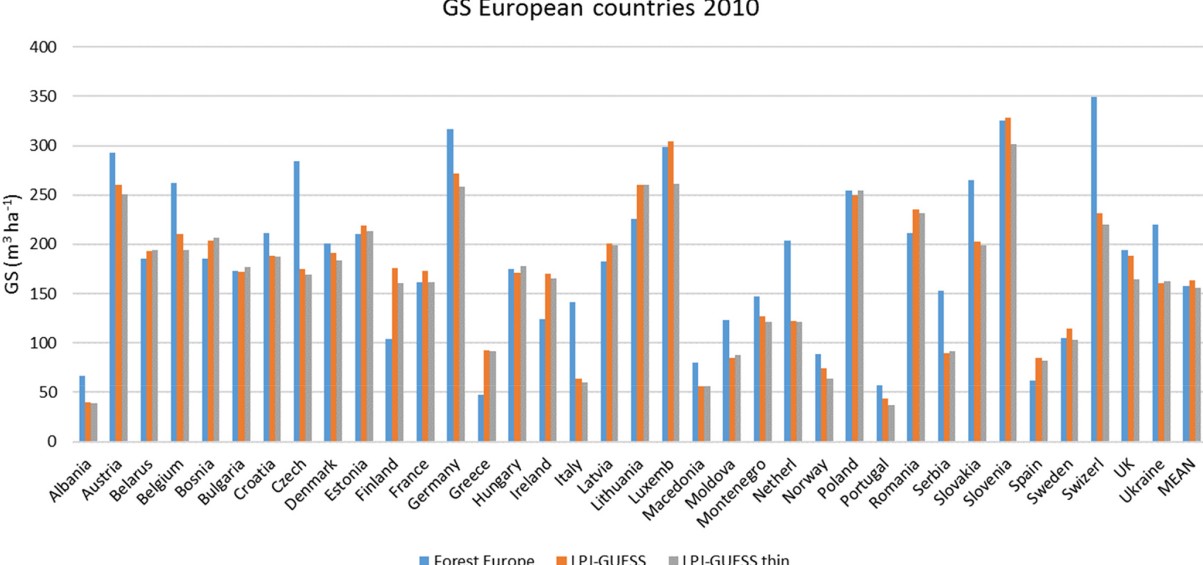

**Figure E3.** Modelled and observed (Forest Europe, 2015) growing stock (GS) for individual countries in 2010. LPJ-GUESS is the simulation without thinning. LPJ-GUESS thin is the simulation with automated thinning.

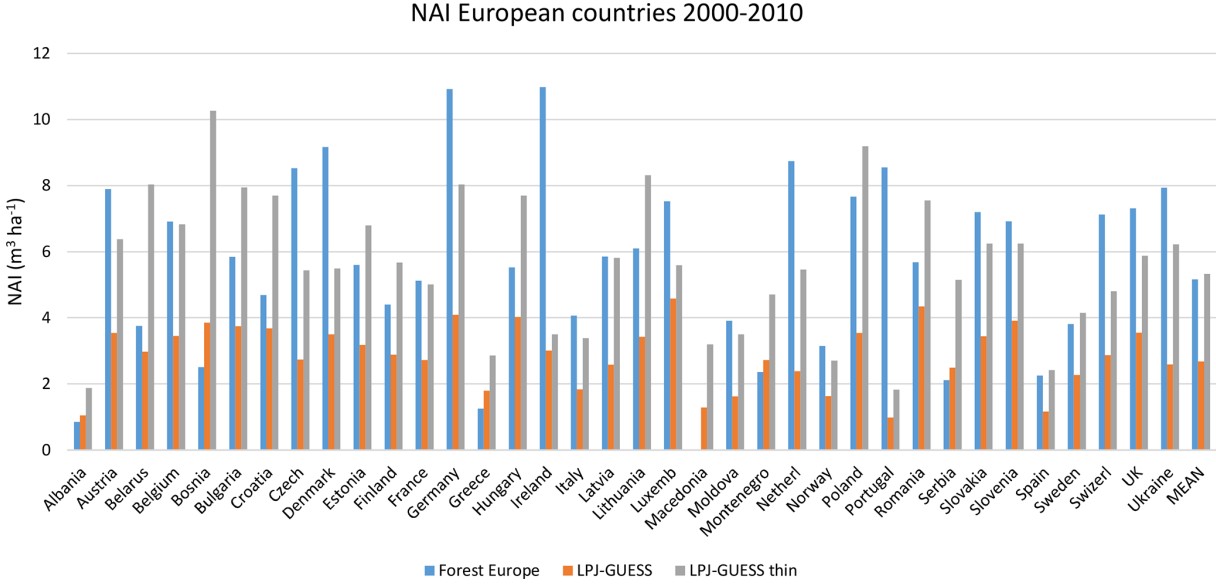

**Figure E4.** Modelled and observed (Forest Europe, 2015) net annual increment (NAI) for individual countries in 2001–2010. LPJ-GUESS is the simulation without thinning. LPJ-GUESS thin is the simulation with automated thinning.

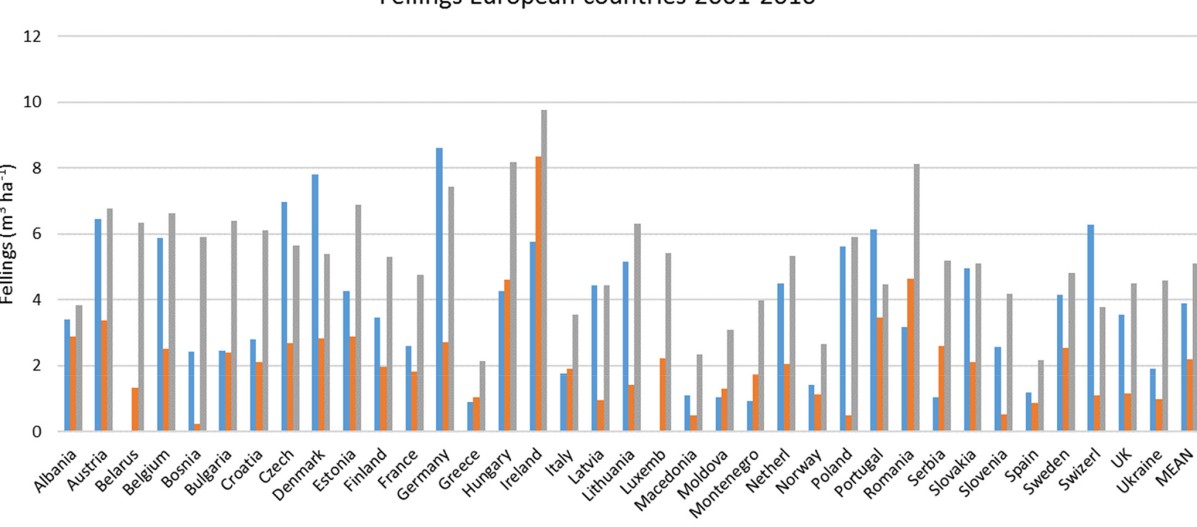

**Figure E5.** Modelled and reported (Forest Europe, 2015) yearly fellings for individual countries in 2001–2010. LPJ-GUESS is the simulation without thinning (clear-cutting at creation of secondary forest). LPJ-GUESS thin is the simulation with automated thinning. Reported values are missing for Belarus and Luxembourg.

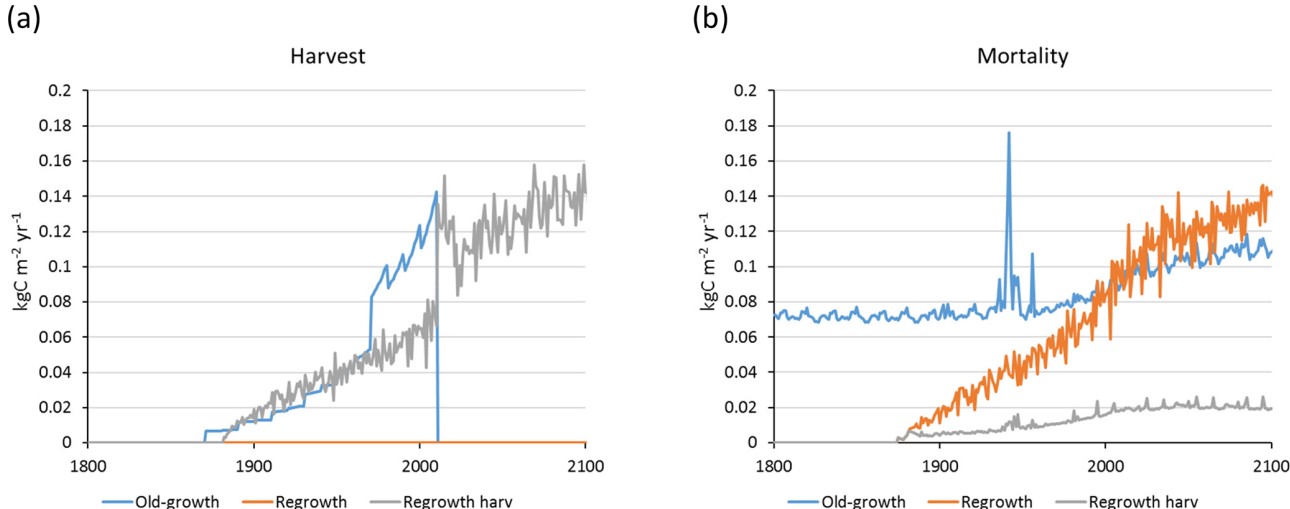

**Figure E6.** Simulation of European old-growth and regrowth forests with (Regrowth harv) and without (Regrowth) wood harvest in regrowth forests using historic CRU-NCEP climate, recycling the last 30 data years after 2015. **(a)** Harvested carbon. Old-growth harvests are clear-cuttings at the creation of secondary (regrowth) stands in the period 1870–2010. The spike in regrowth forest harvest in 2011–2020 is due to delayed clear-cutting of stands passing the tree density limit for clear-cutting before 2010. **(b)** Vegetation carbon lost in natural mortality.

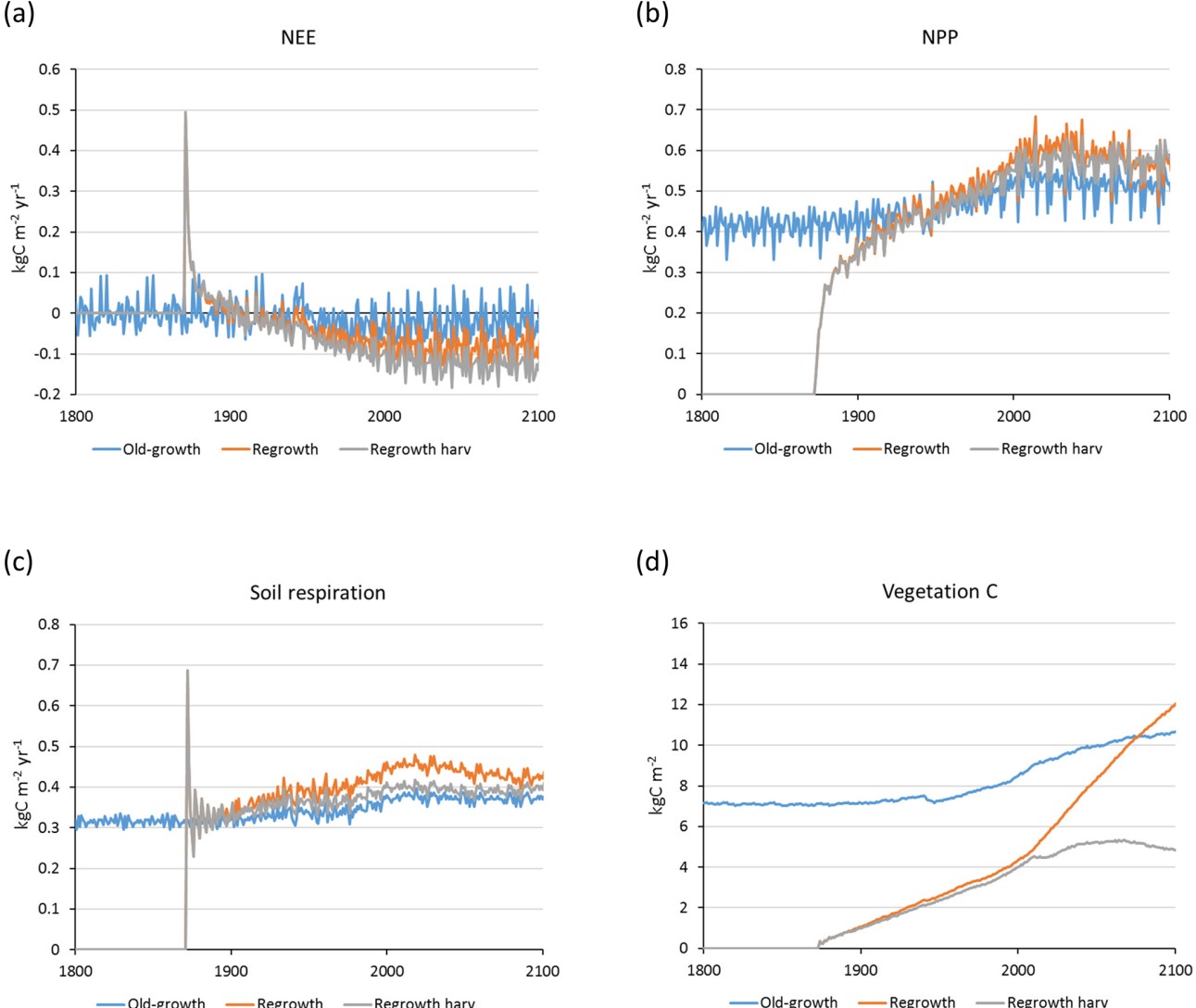

**Figure E7.** Simulation of European old-growth and regrowth forests with and without wood harvest in regrowth forests using historic CRU-NCEP climate, recycling the last 30 data years after 2015: **(a)** net ecosystem exchange (NEE), **(b)** net primary productivity (NPP), **(c)** soil heterotrophic respiration, and **(d)** vegetation carbon. Some NEE components are not shown, e.g. carbon allocated to reproduction and fire in old-growth forest.

(a)

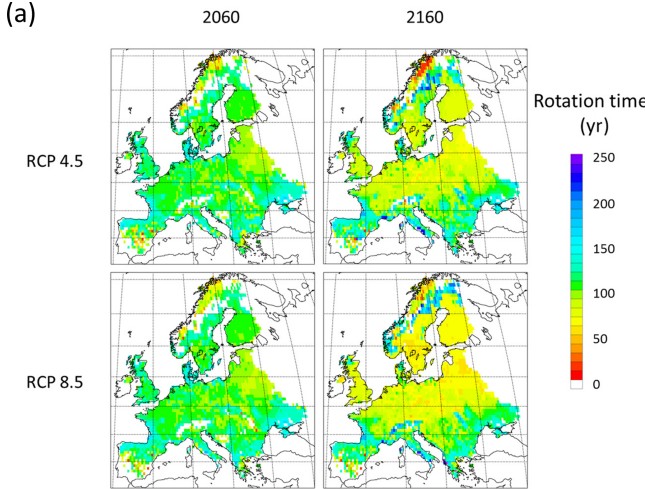

(b)

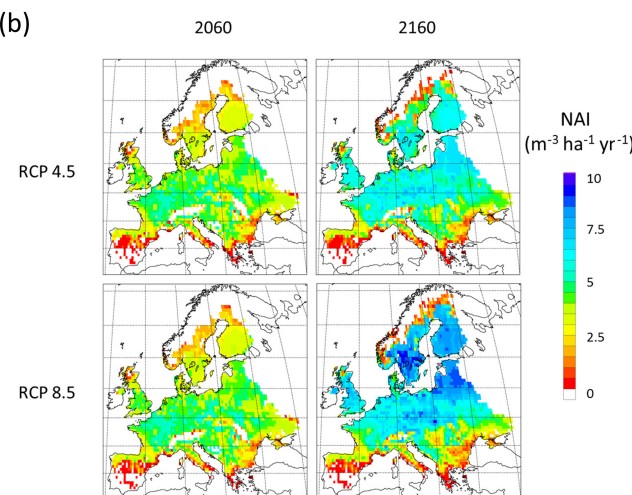

**Figure E8.** Simulations of broadleaf forests using automated thinning and clear-cutting under RCP 4.5 and RCP 8.5 $CO_2$ and climate, recycling the last 30 climate data years after 2100. **(a)** Mean rotation time for the latest clear-cut events in each stand in 2060 and 2160. **(b)** Mean net annual increment (NAI) during the latest rotations in each stand in 2060 and 2160. For the expansion from total vegetation carbon to wood volume, a wood volume/vegetation carbon ratio of 2.7 TS6 $m^3\,t\,C^{-1}$ was used.

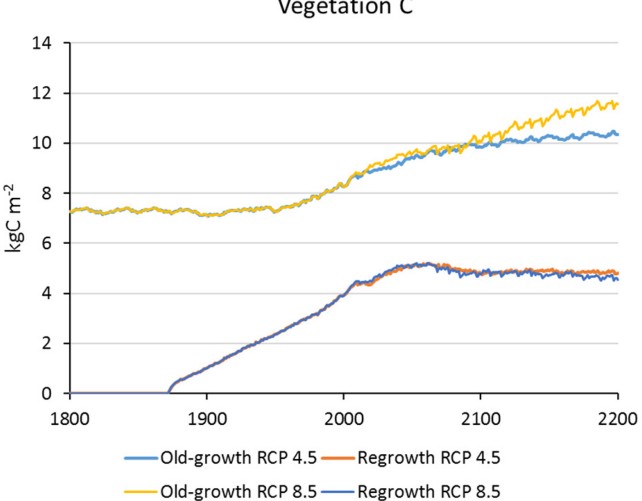

**Figure E9.** Simulations of European forests using automated thinning and clear-cutting in regrowth forests under RCP4.5 and RCP8.5 $CO_2$ and climate, recycling the last 30 data years after 2100. Vegetation carbon in old-growth and regrowth forests. Old-growth forests are simulated as PNV.

*Code availability.* LPJ-GUESS development is managed and the code maintained in a permanent repository at Lund University, Sweden. Source code is normally made available on request to research users. Conditions apply in the case of model versions still under active development. The model version presented in this paper is identified by the permanent revision number r9710 in the code repository. There is no DOI associated with the code.

*Data availability.* Observational and modelled data used to create figures and tables are available at https://doi.org/10.5281/zenodo.5553194 (Lindeskog et al., 2021).

*Author contributions.* ML and FL developed the forestry model code. ML and AR designed the simulations. ML performed the simulations and designed and performed the analyses. HP provided the GER-Bav site data. AF provided the SLO site data. ES contributed to the forest site data simulation setup and analysis. All authors contributed to the manuscript.

*Competing interests.* The authors declare that they have no conflict of interest.

*Acknowledgements.* This study was funded by the Swedish Research Council Formas through the ERA-Net SUMFOREST project Forests and extreme weather events: Solutions for risk resilient management in a changing climate (FOREXCLIM), the project Land Use, Carbon Sinks and Negative Emissions for Climate Targets of the German Federal Office for Agriculture and Food (BLE) through the FOREXCLIM project, and by the Slovenian Ministry of Agriculture, Forestry and Food (MKGP) through the FOREXCLIM project. The study also contributes to the Strategic Research Areas BECC and MERGE. Anja Rammig acknowledges funding from the Bavarian Ministry of Science and the Arts (BayKliF). The computations were enabled by resources provided by the Swedish National Infrastructure for Computing (SNIC) at LUNARC, Lund University, partially funded by the Swedish Research Council. We thank Gerhard Schütze, Martin Nickel, and Leonhard Steinacker for providing measurement data from Bavaria. Further we thank the Bayerische Staatsforsten (BaySF) for providing the observational plots and the Bavarian State Ministry of Food, Agriculture, and Forestry for permanent support of the projectW07 "Long-term experimental plots for forest growth and yield research". We thank Ralf Nagel and the Nordwestdeutsche Forstliche Versuchsanstalt, Göttingen, for providing the measurement data from central Germany. We thank Thomas Pugh for helpful discussions.

*Financial support.* This research has been supported by the Swedish Research Council Formas (grant nos. 2016-02110 and 2016-01201), the Swedish Research Council (grant no. 2019/3-592), the German Federal Office for Agriculture and Food (BLE) (grant no. 2816ERA01S), the Bavarian Ministry of Science and the Arts (BayKliF), the Bavarian State Ministry of Food, Agriculture, and Forestry (grant no. 7831-26625-2017), the Slovenian Ministry of Agriculture, Forestry and Food (MKGP) (grant no. 2330-17-000077) and the Slovenian Research Agency (research core funding grant no. P4-0059).

*Review statement.* This paper was edited by Tomomichi Kato and reviewed by two anonymous referees.

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
