# Peer review of "Accounting for forest management in the estimation of forest carbon balance using the dynamic vegetation model LPJ-GUESS (v4.0, r9710): Implementation and evaluation of simulations for Europe."

_Geoscientific Model Development, 2020_

## Author Response (AR1)

We would like to express our gratitude to the referees for the provision of excellent comments and suggestions for improvement of the manuscript. We hope that we in our reply can envisage sufficient steps for making the manuscript suitable for publication.

Referee #1

*General comments*

*As I think it is urgently necessary to consider more realistic stand conditions as well as management impacts into estimates of carbon and wood storages, I welcome the incorporation of such features into LPG-GUESS. I think that the results obtained for managed forest area are indeed advancing modelling science. Despite being not the first modelling approach addressing biomass development in dependence on forest management, I guess it is worthwhile publishing the respective components specifically for LPJ, in particular since the now available options are quite comprehensive and the question addressed is still not well investigated. In my opinion, the presentation is well structured and the language is fluent (some spelling errors may occur that are not specifically pinpointed but this can easily be corrected). Some specific recommendations related to references, figures and issues that might be better addressed are given below.*

*Two main concerns remain that may require some more elaboration. Firstly, I am a bit irritated about the presentation of large-scale simulations that use the new features at a large scale without presenting any stand-level evaluation of managed sites in the paper. As it is, it is implicitly assumed that if the model represents PFTs under natural conditions correctly, it is also appropriately simulating the PFTs under management. Since it is known that growth depends on structural features (e.g. Bohn et al.) and this is what is changed here, this assumption might not hold, and thus should be demonstrated, at least in case of some examples. In fact, I think that lots of specific cases are available that could be used for such an exercise (e.g. sites in the Profound database). Please apologize in case such evaluations have been presented in other publications and I have overlooked the references.*

REPLY: Stand-level simulations were originally meant to be included in the manuscript, but was left out for space reasons. We will add a figure/table of model performance regarding standing volume and harvested volume in monoculture beech and spruce (space allowing, also pine and oak) stands located in a few European countries. Input to the model will be stand age, tree species and harvest intensity. Harvesting alternatives will be: no harvest, detailed harvest intensity every 5-10 years and automated harvest as used in the European simulations.

[revised manuscript text omitted]

l. 663-665: While mean productivity of European forests is captured well by the model (Table 4) and productivity of forests in individual European countries reasonably well (Fig. 10, Fig. E4), the inability to reproduce observed productivity levels in high-productivity beech and spruce stands in Germany (Fig. 7) highlights the need for allowing a wider range of productivities.

*With this respect, I am particular puzzled about the example simulation, where beech seems to perform the superior growth relative to spruce in the PNV (for the first 100 years) as well as (and particularly in) all managed forest options. This seems to be counter-intuitive to what I would expect for a site in southern Sweden (see e.g. Bolte et al. 2010), although I am aware of only one extensive biomass study which gives almost the same biomass/production values for both species in this area (Nihlgard 1972). However, although the particular conditions seem to be more representative to Poland than to Sweden (see e.g. Jagodizinki et al. 2020), I might misjudge the growth and will happily accept if I am proven wrong. In any case, I think that particularly (the) illustrating example(s) need some data or references that support their realism.*

REPLY: While it is our aim to improve the realism of interspecies competition in the model, we don't expect a DGVM to to a perfect job for every species in every region. Current developments of the model, out of scope for this article, that should be able to achieve a more realistic interspecies competition include an improved soil water-holding capacity description and growth limitations of nutrients other than nitrogen. Another important factor that influences competition at the tree seedling stage is herbivory, which is not yet currently implemented in the model.

At our south Swedish site in Figures 2 and 4, it is true that beech out-competes spruce when growing in PNV at the beginning of a vegetation succession (Fig.2b) and in mixed beech-spruce stands (Fig.3a). However, when grown in monocultures (Fig.4b), beach and spruce grows equally well until 1985 (when measured in kgC/m$^2$), when beech growth increases, and when measured in wood volume, spruce is ahead of beech all the time (ending up at 377 m$^3$/ha vs. 302 m$^3$/ha in 2000). Figure 4 illustrates the two different ways to achieve a pre-defined species mix in the model, but we could add a plot showing the standing volume development of spruce and beech although it might detract from the main message. The fact that the plot inside the red box Fig.4b is the biomass C development at the landscape level and that it's not possible to compare the growth of the two species in that plot can be better stressed in the figure caption.

We will add a short section in the discussion about present problems in interspecies competition between the European species in the model.

Added to Fig.4: (c), showing the development of beech and spruce standing volume in Fig. 4b in their separate stands.

Added to discussion. l. 666-668: The emergent competition between PFT:s with similar shade tolerance values in the model, e.g. beech and spruce, can deviate from actual dynamics, as seen in the poor performance of spruce compared to beech in a succession at the example site in southern Sweden (Fig. 5).

*Second, it is unclear why old forests (one third of the area) are considered as PNV despite it is acknowledged by the authors that this leads to a simulated species composition widely divergent from the actual forests. It also leads to total forest estimates of carbon in the different compartments that are heavily influenced from this (unrealistic?) assumption. I guess that considering even a species composition from coarse data as is available from literature would largely decrease uncertainties related to the inventory as it is now.*

REPLY: The representation of >140y forests as PNV is meant as a conservative compromise and should result in an underestimation of the C sink of European forests. Alternative setups using the GFAD age database would not necessarily improve the situation satisfactorily. Spinup using PNV, attaining a state of equilibrium of vegetation and soil carbon/nitrogen, is the standard procedure in LPJ-GUESS, has been evaluated extensively and has an uncertainty due to lack of land-use history which is to some extent predictable, while a spinup using the four GFAD forest classes would introduce further uncertainties. Thus, since we don't have any information about when to initialise the old-growth forest classes in GFAD and we believe that unsupported assumptions about age structure is more problematic than uncertainties regarding species coverage, we find it far from ideal but less problematic at this stage to leave the old-growth forest as PNV with a disturbance interval of 400 years. This also allows us to compare more directly with the global simulations of Pugh et al. (2020). Also, the >140y forests in GFAD contain disturbing artefacts, most likely as a result of including shrubland as forest. There are large (up to 20 % in eastern Europe) areas supposedly containing broadleaf evergreen forests and far too much needleleaf deciduous forests. The regrowth part of GFAD is not affected by these peculiarities, so in the European simulations, we like to focus on the regrowth part and treat >140y forest as a source for improvement for future setup refinements. We will further stress this in the text.

Also, the errors in NAI, GS and harvested volume resulting from species inconsistencies in >140y forests are expected to be modest since these are calculated from conversion factors for each European country and not from species-specific wood volumes.

Added to Methods 2.6.1, l. 329-330: old-growth forests (> 140 years; denoted as 'old-growth' forest henceforth, not implying pristine forests). l. 334-336: The oldest forest class in GFAD (>140 years) contains artefacts manifested e.g. as BE occurrences in northern Europe, so the forest type information in this part of the dataset was not used.

Added to Results, l. 433-434: … but since old-growth forest is modelled as PNV in this study because of artefacts in the >140 year data (cf. 2.6.1),…

Added to Discussion l. 606-609: … old-growth (>140 year) forests in this study are represented by unmanaged PNV (with a low carbon sink, cf. Table 6), as in Pugh et al. (2019), missing effects of land-use history in Europe, but preferred by us to the alternative of introducing arbitrary assumptions of age structure. Furthermore, the GFAD >140 year forest type data contain artefacts manifested in the BE distribution.

*Specific comments*

*L17/18: How is the underestimation of carbon sink explained if stock and net growth is fairly well met?*

REPLY: Two obvious reasons for this are a too-high soil carbon content because of a lack of land-use history in the setup that is leaking into the atmosphere after the creation of secondary stands, as well as an unrealistic lack of any removal of biomass in the >140y forests (see reply about this and suggestions for improvement to referee #2). We can add a short sentence about this in the abstract.

Added to Abstract, l. 24-25: …most likely reflecting uncertainties in carbon fluxes from soil given the unaccounted-for forest land use history in the simulations.

*L29: Morin et al. 2017 not in ref. Do you mean Morin et al. 2018?*

REPLY: Yes, thank you for spotting this error.

Corrected reference

*L41: Are the Pan et al. data really providing below- and aboveground carbon stocks for Europe? The numbers given here don't seem to match the same values indicated for the same source in Table 2. Estimates from other sources seem to be also far different as e.g. app. 5/5 (Liski et al.), 9/14 (Goodale et al.), and 9/8 PgC (Pilli et al.). Please check. Consider to introduce the given references into Table 2-4 if appropriate.*

REPLY: Table S3 in Pan et al. 2011 provides carbon in total living biomass, dead wood, litter and soil. We included dead wood in the soil/litter compartment.We will add this detail in the table caption. We didn't manage to find discrepancies between L41 and table 3:  L41: '..store approximately 13 PgC in vegetation and 28 PgC in soils (Pan et al., 2011)', Table 3, year 2007: VegC 13 Pg, Soil+Litter 28.0 Pg (Pan et al., 2011).

The suggested references seem to include less than complete coverages of the study area (e.g. western Europe, excluding baltic countries, Belarus and Ukraine). Pan et al. includes data for soil and litter for all european countries, even though they have to rely on estimates from similar countries in 14 cases. The vakues also agree well with the other source of comparison in the manuscript, Forest Europe, taken into the account that soil data is missing there for a number of countries. We believe the message comes through even if the references used claim higher soil pools than others, i.e that the simulation setup probably overestimates soil carbon.

Added to Table 3: "Litter includes dead wood"

*L69ff: It is not clear, what are newly developed features that justify this publication and what are features that are only described in order to understand the scenarios modelled. Please rephrase and better indicate the expected benefit of the model improvement. (In other words: clarify the objective.)*

Added to Introduction, l. 77-82: In this study, we describe the implementation of expanded forest management capabilities including even-age/clear-cut and uneven-age/continuous-cover management in LPJ-GUESS v.4.0. In addition to detailed carbon- and water-cycle processes, this version of the model incorporates a dynamic nitrogen cycle and nitrogen limitation on plant productivity (Smith et al., 2014). With this, forest management in LPJ-GUESS is for the first time fully integrated in a model version

capable of simulating a landscape containing a mosaic of land cover types like PNV, cropland, pasture and peatland and with a sophisticated land-use and land-cover change functionality.

*L76ff: What is (are) the timestep(s) of the model?*

Added to Methods 2.1, l. 95-96: Photosynthesis, respiration, phenology, soil carbon and nitrogen cycling and hydrology occur at a daily time step, while biomass growth allocation and turnover, establishment and mortality occur at a yearly time step.

*L132ff: For the within-patch mixture of Fig.4a, the 60/40 ratio refers to while for the among-stand-types mixture of Fig. 4b the 60/40 ratio refers to groundcover area. I think that this should be indicated when presenting the different options.*

REPLY: We agree, this will be stressed more clearly in the 2.2.3 text and in the Figure 4 caption.

Added to 2.2.3, l. 157-158: relative biomass abundances…, predefined groundcover area-based mixtures

Fig 4. caption: (a) … (target cutting to a 60/40 % biomass ratio), (b) … patch created in 1901 with 60 % and 40 % groundcover area fractions

*L163: define LUH2*

Added to 2.3.1, l. 198-199: with reconstructed time series of land use from the Land Use Harmonization Project (LUH2, Hurtt et al., 2017)

Added Hurtt et al. reference

*L192/Tab. 1: consider changing 'N fertilization' to 'N fertilization/deposition' since the continuous kind of application looks more similar to a deposition regime (and is more likely to happen anyway).*

REPLY: Nitrogen fertilisation, even though not currently available at defined points in time in the model, can be applied per management type on top of N deposition, which is applied as standard in all simulations, as described in the Methods. Although N leaching will be unrealistic, extra N will increase growth in the model considerably e.g. in Scandinavia.

*L205/Fig. 5: This is a nice presentation indicating the potential options that can be selected. However, I wonder what happens, if 'old' would be selected together with 'small' or vice versa in the thinning rule options? What would happen? Wouldn't be a preference 'from above' or 'from below' be more suitable? What effect does this have on the simulation if biomass is taken e.g. preferentially from larger trees? I guess that the effect would depend on underlying assumptions about the biomass in different size groups – what are these assumptions?*

REPLY: Size preference overrides age preference, so there are no ambiguities possible. Age or size selection are most often interchangable in the model, but not always. But the terms 'from above' and 'from below' could be helpful to mention in the text or in the table. We will try to incorporate these terms. Since the cohorts suffer bioth intra- and interspecies competition for light in the layered canopy, the choice of cutting from below or above will be important for the success of the different cohorts and especially for the amount of regeneration of seedlings over time. The biomass in different size groups is a dynamic outcome of competition beween cohorts and species.

Added to 2.3.2.2, l. 228:  In (1) and (2), size overrides age settings

Added to 2.3.2.2, l. 224-225 ("from above") and ("from below")

*L215ff: I am a bit unhappy with the unit 'densmax' being trees per ha since it implies that the stand is harvested after a certain number of trees has reached. However, the number is actually not defined and can be very different dependent on the diameter distribution. Densmax thus is an arbitrary density value in 1/m2 or similar unit. Similarly, I guess that 'dens' is also not indicating real tree numbers.*

REPLY: The terms in the equations are exactly as in Bellassen et al. We think we should use the same terms to avoid confusion. The different cohorts of a pft are represented by a mean individual with a density in trees per $m^2$.

*L232/233: As indicated in Tab. 1, irrigation seems to simply bypass drought stress. Here it is said that you actually calculate the (minimum) amount of water that is necessary to do this. Is this true? Check and homogenize.*

REPLY: The text is correct. Table 1 entry changed to 'water amount required to avoid water stress in photosynthesis added to soil'

*L308: harvested volume was calculated from 'killed vegetation carbon'? consider rewording.*

REPLY: Agreed, this sounds akward. We should be able to use the wording 'harvest fraction' instead (since we already mentioned vegetation carbon).

Changed text in 2.6.3, l. 361 to: Growing stock, net annual increment (NAI) and harvested volume were calculated from vegetation carbon, net ecosystem exchange (NEE) and total carbon of harvested trees, respectively,…

*L311: What does 'reductions in wood products and residuals' mean? Do you mean 'due to' instead of 'in'? Consider that the argument is repeated a couple of lines later.*

Changed text in 2.6.3, l. 365: not taking into account the fate of wood products and residues following removal from the site

*L330-332: irritating punctuation (needs ":" after forest, and "," after pubescens)*

REPLY: Revised text in l. 380-381 as suggested.

*L486: The simulations are initialized and driven by the given information but not 'constrained', correct? The word would be correct, if repeated inventory data would be used to parameterize or adjust the simulations which I think has not been done.*

REPLY: If using groundcover area and land-cover change functionality, we can constrain the simulations- In other cases, we can only constrain relative species mixes and harvest fractions, so yes, the wording is not accurate.

Changed text in Discussion, l. 579-581: The initialisation and harvest alternatives in the model are tailored to enable available forest inventory data and harvest information to be used to initialise and guide simulations.

*L494: As I take it, there are no observations but only estimates of 'mean growing stock' that are based on different information, sometimes on rather inhomogeneous sources, correct?*

REPLY: That is probably true. We can change 'observations' to 'reported values' in most places in the text.

Changed: 'observations' to 'reported values' at 8 places

*L526: You mean that the NPP decline that ORCHIDEE simulates is not simulated by LPJ, which might have various reasons that are caused by the model structure and processes. You don't mean that the decline is a function as such that is 'included' or not 'included', correct? Could such a decline be related to nutrient depletion? How is the effect of nutrient export – that is a main concern of a sustainable management – considered anyway? I think it should at least be part of the discussion (see e.g. Parolari and Porporato; Sverdrup et al.)*

REPLY: In some earlier versions of our forest management code, a reduction factor was applied to the light extiction coefficient after thinning, following Näslund et al. (1971, Stud.For.Suec. 89:124) and Hale (2003, For.Ecol.Manage. 179:341-349). This functionality was not included in the current version. But the sentence omits a lot of information and we are uncertain about by which mechanism this decline is achieved in ORCHIDEE, so we will change the wording to 'is not simulated in this version of LPJ-GUESS'.

Changed text in Discussion, l. 645: "is not simulated by LPJ-GUESS"

Nitrogen depletion of the soil in previous land-use history has an effect on forest growth in the model as seen in Figure 2c. As discussed in the reply to referee #2's comment on line 107, a change in the amount of removed leaf nitrogen causes little changes in productivity and carbon pools at the European scale using the current version of LPJ-GUESS, but in further studies, the effect of different management practices on productivity at both stand and regional scale will be fully investigated. Obviously, the effect of nutrient export in different management schemes is potentially a very important use of forest management in LPJ-GUESS. Upcoming updates to the model such as improved soil water holding capacity description, growth limitation and cycling of nutrient other than nitrogen, will hopefully improve the usefulness of the model in this respect.

We will add a part of this to the discussion after consulting the suggested references.

Added to Discussion l. 651-662 Responses of soil carbon and nitrogen cycling to harvest and fertilisation can be complex and qualitatively different in clear-cut and continuous-harvest systems (Parolari et al. 2016). The coupled carbon-nitrogen cycling in LPJ-GUESS (Smith et al. 2014) should enable the investigation of the effect of different management practices on forest productivity and sustainability at both stand and regional scale in future studies. Nitrogen depletion of the soil in previous land-use history reduces forest productivity and causes a shift in species succession in the model (Fig. 2c). At the European scale, removing a smaller fraction of residues (0 % of leaves rather than 30 %) makes a small positive impact on productivity (0.1%, cf. 3.4). However, since many European forests receive large amounts of atmospheric nitrogen deposition, other nutrients such as Ca, Mg, K and P may be more important for limiting productivity, and acidification of the soil by N and S deposition may further decrease the availability of these nutrients (Sverdrup et al. 2006.). Especially Ca is close to or below the limit of sustainability in current forest management systems in southern Sweden (Sverdrup et al. 2006). Thus, ongoing development of limitation and cycling of additional nutrient species into LPJ-GUESS may be beneficial for capturing the full effects of different harvest regimes. Also relevant to achieving a better model of nutrient uptake is an improved representation of the soil profile.

*L645: I don't see the usefulness of this plot. The development of stand density in dependence on automated or self-thinning seems to be the same for all countries as is not surprising since it is treated with the same functions. So what would you like to show here?*

We considered removing this figure, but reconsidered this since it is essential to explaining why automated thinning for beech does not behave exactly as expected from the self-thinning rule. Furthermore, we believe that self-thinning is not identical for all locations and species, since it is the outcome of climate, stand structure and life history.

Referee #2:

*Including a more realistic forest management in DGVMs is important, and this manuscript clearly addresses and demonstrates this. Although everything is presented and explained, it was difficult to follow all approaches and concepts in the first read, perhaps because I am not familiar with LPJ-GUESS. Including (many) different options to achieve the same result, and extensively listing alternative options or exceptions are confusing and distract from the main message. An example of the first case is the presentation of 3 alternative ways of creating an (initial) age class distribution (Section 2.2.2., Figure 2), which requires already quite a good understanding of the concepts on patch/stand/management type in LPJ-GUESS. An example of the second case is section 2.3.2.1 (Species selection). It is full of words like "may", "can", "or", "optionally" and "possibly".*

*I think it would be worthwhile to introduce early on in the paper predefined categories, and explain how these are implemented in LPJ-GUESS. For Europe, I would say three types of management are important to consider: age-class (even-aged) based forestry, uneven-aged forestry/continuous cover forestry, and pristine forests. Pristine forests would be implemented as Natural (PNV) in Figure 1, while even-aged and uneven-aged management would have a separate stand type/management type. Even-aged forestry would logically be represented by having stands of different age, with patches of the same age. Uneven-aged forestry would then be represented by one stand, containing patches of different ages.*

*Similarly, in Section 2.3 (Forest management routines), it would be very helpful to announce which systems will be covered in this section (simplified forestry, detailed forestry and continuous cutting), and connect this to the management types defined earlier. Also, I think it makes sense to present Table 1 already here in the introduction to this section. Table 1 could be extended with columns that indicate for the different systems (simplified, detailed, continuous) the options implemented. Please see the example attached.*

REPLY: The suggestions of the referee concerning the presentation of the different management methods are all useful. We will add more background why the different options are needed, structure the text within the frame of the three major types of management and revise Table 1 according to the suggestions.

Added to Abstract l. 15: …forest management module containing even-age/clear-cut and uneven-age/continuous-cover management alternatives

Added to Introduction l. 77-78: In this study, we describe the implementation of expanded forest management capabilities including even-age/clear-cut and uneven-age/continuous-cover management in LPJ-GUESS v.4.0.

Added to Methods 2.1 l. 127-131: The typical forest management types covered in the model and presented in this paper are: no management (pristine forests, simulated as PNV), even-aged forestry, typically modelled by stands with prescribed ages starting from bare ground after a specified land-use history, and uneven-aged/continuous cover forestry, typically modelled by a cohort structure within a patch derived from prescribed cuttings after starting as bare ground and a regeneration phase. Alternatives to these typical setups can be used to achieve age structures at other spatial scales, e.g. landscape level and will be described below.

Added to Methods 2.3 l. 180-181: Two types of harvest systems are available in the model: clear-cutting and continuous cutting, which are used in conjunction with the even-aged and uneven-aged/continuous-cover age-structure systems, respectively (Table 1).

Changed Table 1

*The authors use the term "old-growth" forests for forests that have a certain age. However, in the literature old-growth forests is commonly used as a synonym for pristine forests, and the authors should avoid this confusion by using another term. The assumption that forests older than 140 years of age are not managed is not totally realistic, and the assumption that these have a species distribution equal to PNV is not realistic at all. I see no problem in applying assigning species to this class in the same way as for the younger age classes. Rather than assuming no management, these forests could be managed with low intensity using the continuous cover forests management type.*

REPLY: In the context of the European simulations, the definition of "old-growth" forests are aligned with the corresponding GFAD-driven global simulations in Pugh et al. (2019). We would like to keep the same terminology as in that publication, realising the risk that this could cause some confusion. We will more clearly define the term as used in the text where necessary.

Regarding the assigningment the >140yforest to the forest classes in GFAD, please see our reply on the matter to referee #1.

The lack of harvest in the >140y forest is more difficult to defend. The model doesn't currently provide an automated continuous cutting method that allows for realistic regeneration of seedlings. We are currently studying the effect of continous harvest intensity and timing in mixed forests and hope to be able to introduce a working method in the near future. For this study, we believe a sensible alternative to be the removal of wood products (as in normal wood harvest in the model) at the patch-destroying disturbance events (at an expected frequency of 1/400 years) that are present in the >140y forests simulated as PNV. We propose to include this as an alternative setup for the "old-growth" forest. This alternative results in small changes in biomass (+0.03 PgC; <0.1% for the total area, ca. 0.15% for the >140y area), modest but significant changes in soil+litter C (-0.6 PgC; 1.3% for the total area, ca. 2.4% for the >140y area) in 2010, and larger changes for the carbon sink (+0.016 PgC/y; 6.8% for the total area, ca. 35% for the >140y area) in 2001-2010. This will increase the C sink in 2001-2010 from 0.075 kgC/m$^2$/y to 0.079 kgC/m$^2$/y.

Added to Methods 2.6.2, l. 356-358: To perform a limited sensitivity test of some of the uncertainties in land-use and residue removal assumptions, additional alternative simulations were performed: a simulation where a fraction (as in standard harvest) of the biomass of killed trees in disturbance events in >140 year forests was removed from year 1871, simulating an extensive wood harvest scheme;

Added to Results 3.4, l. 513-517: In a simulation with removal of biomass during disturbance events in the old-growth stands (not shown), the carbon sink in this forest class increased to 0.04 PgC y-1 or 0.05 kgC m-2 y-1 in 2001-2010 compared to a standard simulation, increasing the total forest carbon sink in the same period by 7% to 0.25 PgC y-1. Soil/litter carbon in the old-growth forest was reduced by 2.4% in 2010 and by 0.7% in the regrowth forest, reducing the total soil/litter pool by 1.3%. Total vegetation carbon increased by 0.24% in 2010.

Added to Discussion, l. 609-610: Including a basic extensive wood harvest method in old-growth forests would be expected to increased the total carbon sink by only 4 %., resulting in a value of 66 % of the Pan et al. (2011) value.

*Line 107 mentions that 30% of leaf biomass is removed from the site in case of harvesting. This is way too high. Delimbing of felled trees is mostly done in the forest and in that case all branches and leaves remain in the forest. In case of harvest residue extraction, branches are left on piles for a while to dry out and drop the foliage. These piles may be at the roadside, not sure that counts as being in the forest or not. Foresters like to keep the foliage in the forest for nutrient recycling, and biomass plants prefer having as little as possible foliage. Harvest residue extraction is only applied at large scale in part of Europe's forests (mainly Nordic countries). I would estimate this value at 5-10%. A sensitivity analysis on this number would be very valuable.*

REPLY: The figure of 30% is based on figures that make sense for conifers in Sweden. Nutrients in branches in roadside piles should probably be excluded from the soil in the forest. as they are in the model. Using leaf removal amounts that are realistic for only part of Europe's forests is obviously not a preferred solution, but in the context of other uncertainties in the forest management setup (which will be addressed in a specialised study), the expected effects of changing leaf removal amount are small. However, to show that this is the case, we performed European simulations with 0 and 10% leaf removal. Changes for the 0%/10% options were: vegetation C +0.13%/0.08%, soil+litter C +0.10%/0.07% in 2010, C sink -0.003/0.002 PgC/y (1.3%/0.89%). It is possible that improvements of the model (soil, nutrients other than N) might increase the sensitivity to biomass removal options.

Added to Methods 2.6.2, l. 356-359: To perform a limited sensitivity test of some of the uncertainties in land-use and residue removal assumptions, additional alternative simulations were performed: … ; two simulations where the leaf removal fraction in harvest events was set to 10% and 0%, respectively, instead of the standard 30% value.

Added to Results 3.4, l. 518-521: Simulations with alternative settings of leaf removal fractions during harvests of 10% or 0%, instead of 30% in the standard simulation (not shown), decreased the total carbon sink in 2001-2010 by 0.9% and 1.3%, respectively, resulting from an increased soil respiration of 0.3% and 0.4% respectively, partially offset by an increase in NPP by 0.06% and 0.09%, respectively. Vegetation carbon increased by 0.08% and 0.13% and soil/litter carbon increased by 0.07% and 0.10% in these simulations.

Added to Discussion, l. 655-656: At the European scale, removing a smaller fraction of residues (0% of leaves rather than 30%) makes a small positive impact on productivity (0.1%, cf. 3.4).

*Line 200: the minimum diameter for harvesting can be lowered if no trees are available. I wonder if this is realistic. Harvesting small trees is costly and the revenue is small. In such cases I would think harvesting is simply not done at all.*

REPLY: Perhaps this should be treated as an option for test purposes only and not be mentioned in the text. We will remove the option from the text in 2.3.2.3.

Removed text from 2.3.2.3

---

## Author Response (AR2)

Dear Dr. Kato.

Please consider our revised manuscript and response to referee #1 for review. We thank both referees for taking the time to review our manuscript.

Best regards,

Mats Lindeskog

Response to referee #1.

General comments

I appreciate the added exercise of evaluation which certainly have taken some effort to integrate and is necessary in order to give the study some credibility. Nevertheless, based on the results of this exercise, I think that in particular the underestimation of wood volume development for quite some sites should be better discussed (Model sensitivity to environment? Importance of structural issues not considered? Wood density or stem form?). Also, I am a bit irritated that this underestimation doesn't lead to a corresponding underestimation at the regional scale. For the general picture it would thus be good to elaborate a bit on this point (Unluckily selected examples? Problems with the inventories? Importance of disturbances?).

Adding text to Discussion: "The lack of certain physiological processes in the model, e.g. hardening/dehardening (Bergh et al. 1998), could explain why productivities along the whole temperature gradient in European forests cannot be fully represented in the model. Model tuning that aims for correct mean values of e.g. biomass and carbon fluxes over large geographic areas compensates for an overestimation of productivity in northern Europe by lowering average productivity along the whole temperature gradient. This could partially explain e.g. why the productivity of some south German sites is underestimated, while average productivity for Germany as a whole is in line with inventory data. Additionally, the selected individual German Norway spruce and European beech sites in this study were generally of above average site quality, and are not fully representative of German forests, which includes forests of other tree species, especially Scots pine (Pinus sylvestris) and oak species (Quercus robur/Quercus petraea), on lower site-quality sites. This is likewise in line with the smaller gap between modelled and observed growing stock (ca. 20%, Fig. E3) seen at country level, compared to individual spruce and beech sites in Germany (Fig. 7a)."

Adding reference: Bergh, J., McMurtrie, R.E., and Linder, S.: Climatic factors controlling the productivity of Norway spruce: A model-based analysis, For. Ecol. Manag., 110, 127-139, 1998.

I can follow the authors in their argumentation which defends the initialization with old-growth forests based on simulated natural vegetation. I had the impression that arguments and also disadvantages of the one or other possible choice are well balanced.

From a technical point – and acknowledging that a number of smaller errors have already been removed - I feel that in still some places, grammar and wording need to be elaborated in order to increase understandability. Apart from these relatively small issues, I am looking forward to an interesting publication representing a step forward in regional to global modelling.

Specific comments (some issues that caught my eye, not a complete list)

L123 (figure caption): What does 'Stands belonging to stand types with trees can only be reduced in size.' Stands can get smaller? Du you mean patch size? Please find a more indicative wording.

To avoid ambiguities, we changed the sentence to "During land-cover change events, stands belonging to forest stand types can only be reduced in size." This should be in line with the text at the right side of Figure 1 which states that both stand types and stands occupy a dynamic gridcell area fraction, the text on l. 126 "Transitions between different stand types may occur at any point in time" and the text on l. 127-129 "When a potentially forested stand type area expands, new stands are created, keeping the soil history from the previous stand type intact and allowing vegetation succession to proceed from bare ground (in most cases, but cf. 2.2.1)." and hopefully be concise and clear.

L127: ', to recreate land-use history or effect a future land use scenario'. I know what you mean but I think it is not clearly nor correctly written – and may be deleted without any harm.

Changed sentence to "… to take into account land-use history or future land use scenarios."

L157ff: awkward sentence, rephrase and possibly shorten.

The sentence was rephrased and shortened: "To achieve an age structure among patches within a stand, the semi-randomised age structure of PNV (see Section 2.1) may be conserved after the conversion to managed forest if the cloning functionality is used (Fig. B1)."

L161: could you clarify '… may be clear-cut successively at regular intervals'? at one plot 'clearcut' and 'successively' would exclude each other. Do you mean plot by plot or do you mean thinning?

Changed sentence to: "may be clear-cut successively, one by one, at regular intervals", hopefully making it clearer.

L331ff: The sentences are too long and partly confused. In addition, wording and punctuation seems to not precise in places (e.g. setup was done according to specific observations for each place but probably not differently otherwise; wouldn't 'some regeneration' occur independent of the homogeneity of thinning – and isn't that contradicting the later sentence that thinning from below was carried out?). So please check and reword the whole paragraph.

Changed section; splitting sentences and improving wording in some places:

"Four data sets of European beech and Norway spruce monoculture stand time series (1-21 points in time) of standing volume and harvested volume were used in simulations to initialise species and age structure, assuming a landscape distribution of even-aged stands. The stands were located in central and southern Germany (GER-Bav, GER-C, GER-CS) and northern Slovenia (SLO, beech only) (Appendix D, Table D1). Model setup and input climate data were as described in 2.4. Three different harvest strategies were used: no harvest, detailed harvest from observations and automated thinning and clear-cutting (2.3.2). The setup of the detailed harvest for stands from the different data sets differed slightly, depending on the number of harvest data points. For the stands from the GER Bav, GER-C and SLO data sources (3-21 data points per stand), the harvest data (fraction of biomass) were used in the simulations at the reported timings. During the time period prior to the first harvest data point, mean harvest intensities from the harvest data were used, in the case of GER-Bav and GER-C converted to fit a 5-year

harvest interval, while in the case of SLO keeping the 10-year intervals used in the sampling. The GER-CS data contain only one harvest data point for the whole stand lifetime (100 years). In this case, harvests were performed at 5-year intervals during the whole simulation using the calibrated harvest intensity values required to obtain a cumulative harvest fraction equal to the reported harvest fraction for the whole 100-year period. Thinnings in the detailed harvest simulations were performed equally for the different cohorts to obtain some regeneration of saplings in old stands. The automated thinning and clear-cutting method used the parameter settings in Table A3 and thinnings from below started at a stand age of 10 years."

With thinning from below, new saplings are usually removed, at least with a 5-year thinning interval (same as the establishment interval in the model). Some of the German and Slovenian stands are old enough to be experiencing some age-related mortality in the model, so we get slightly less of a biomass decline when thinning all cohorts equally. This is for the detailed harvest simulations and thus not contradicting the thinning from below in the automated thinning simulations, where the tree density-induced clearcut avoids most age-dependent mortality.

L354, 395, 465: 'oblast' is Russian, better use the expressions county or district here.

Changed to "region"

L383: I would appreciate when another expression than 'killed trees' could be found.

Changed from "killed trees in disturbance events" to: "trees killed in natural disturbance events"

L673: Countries have modelled thinning fractions? What does this mean?

Changed sentence: "However, it is obvious that modelled thinning intensities for countries in the Balkans, except Albania and Greece, are higher than the corresponding reported total harvest intensities."